# Position: Artificial Intelligence Needs Meta Intelligence – the Case for Metacognitive AI

**Sergei Chuprov** [1]  **Richard D. Lange** [2]  **Leon Reznik** [2]  **Paulo Shakarian** [3]  **Raman Zatsarenko** [2]
**Dmitrii Korobeinikov** [2]

## Abstract

This position paper argues for metacognition as a general design principle for creating more accurate, secure, and efficient AI. The metacognitive solution involves systems monitoring their own states and judiciously allocating resources depending on each problem instance's difficulty or cost of mistakes. Drawing inspiration both from past work on resource-rational AI and from well-documented metacognitive strategies in psychology and cognitive science, we identify specific challenges in embedding these strategies into AI design and highlight open theoretical and implementation problems. We showcase these principles through a tangible example of improved learning efficiency, effectiveness, and security in a Federated Learning case study. We show how these principles can be translated into practice with a novel software framework developed specifically to allow the community to design, deploy, and experiment with metacognition-enabled AI applications.

## 1. Introduction

Recent advances in both inference and training have incurred significant cost in terms of computational complexity. For example, large language model (LLM) "reasoning models" such as OpenAI 5.x and Claude Opus 4.5 have significantly increased costs at inference time (Wiggers, 2025). Training these models has also become resource-intensive, leading researchers to explore various methods to reduce such costs (Wang et al., 2022). Though current AI systems

exhibit remarkable performance in diverse application areas, the methods for determining if adequate computation has been applied for inference or training are surprisingly rudimentary. Granted, these methods have led to significant advances in the past decade, especially in cases where inference cost is negligible and training is conducted on limited datasets. However, large-scale and widely-deployed models strain available resources both during training and during inference, all the while providing few guarantees on correctness or robustness. In short, there is a strong need for AI systems to monitor and self-regulate their own trade-offs between resource consumption and performance.

Humans, on the other hand, are well known to exert control over their cognitive resources. We make decisions on how long to study on a test (Undorf & Ackerman, 2017), how much time we allocate to a reasoning task (Toplak et al., 2014), and when to request help on a problem (Undorf et al., 2021). These observations have been rigorously studied in cognitive psychology on the topic of *metacognition* (Hen-mon, 1911; Flavell, 1979; Nelson & Narens, 1990; Metcalfe & Shimamura, 1994). In (Ackerman & Thompson, 2017) this concept is defined as "the processes that monitor our ongoing thought processes and control of the allocation of mental resources." Importantly, mere self-monitoring and error-correction is not enough; *post hoc* error-correction adds significant costs to a system, whereas metacognitive monitoring allocates resources up front to navigate trade-offs between cost and performance.

**The position of this paper is that metacognition, encompassing self-monitoring and resource-allocation inspired by cognitive science, should be a general design principle for AI systems. Further, we call for a theoretical framework for metacognition to unify current efforts, and for actionable strategies to translate metacognition research into practical AI applications.**

This paper argues that integrating metacognitive capabilities has significant potential to advance AI, but major challenges lie in their lack of theoretical guarantees as well as difficulties with technical implementation and integration. While most AI research focuses on specific cognitive functions, a meta-level approach enables the dynamic selection of

---

[1]University of Texas Rio Grande Valley, Edinburg, TX, USA. [2]Rochester Institute of Technology, Rochester, NY, USA. [3]Syracuse University, Syracuse, NY, USA.. Correspondence to: Sergei Chuprov <sergei.chuprov@utrgv.edu>, Leon Reznik <leon.reznik@rit.edu>.

*Proceedings of the 43$^{rd}$ International Conference on Machine Learning*, Seoul, South Korea. PMLR 306, 2026. Copyright 2026 by the author(s).

strategies based on their cost and effectiveness. We take inspiration from metacognitive concepts from cognitive psychology (sec. 2) and frame our position against alternative paradigms (sec. 3). We then discuss the application of these concepts in AI (sec. 4), identifying potential functionalities (sec. 2) and demonstrating implementation through case studies (sec. 5). We present a call to action to help the community prioritize the research tasks necessary for advancing metacognitive systems (sec. 6).

**Why this topic is important to discuss at this conference.** This topic will attract the attention and participation of the members of various disciplines expected at ICML from cognitive science to Machine Learning (ML) experts and application developers. We anticipate fruitful collaborations emerging from the discussion of both theoretical and practical aspects of the problem. A discussion on metacognition may reveal pathways to streamlined and more efficient AI/ML development which will not only save costs, but also allow smaller research groups and companies to compete. It will assist in defining major research directions for building safer and more secure, as well as more effective and efficient AI solutions. Furthermore, we note that metacognition has become an area of recent interest to the AI community (Shakarian & Bastian, 2025; Shakarian & Wei, 2025; Didolkar et al., 2024; Leonard et al., 2024; Tankelevitch et al., 2024; Walker et al., 2025; Johnson, 2022; Johnson et al., 2026; Shakarian, 2026). Others have also recently argued for metacognition as a general AI principle (Shakarian, 2026; Wei et al., 2024; Johnson et al., 2026) but our position is unique in highlighting specific open problems and applications related to complexity, efficiency, learning, and security.

## 2. The Case for Metacognitive AI

**Technical Epitomes from Cognitive Science.** Cognitive psychology and related fields provide robust frameworks for metacognition, and we argue that more of these concepts should be applied to AI design. A foundational example is the bi-level paradigm of Nelson and Narens (Nelson & Narens, 1990), where cognition consists of *object-level* and *meta-level* processes. Object-level processes include tasks such as perception, learning, reasoning, and planning, while meta-level processes monitor and assess the object-level processes. While most work done by AI systems today is focused on the object-level, some lines address aspects of the meta-level. One way to further conceptualize these processes stems from dual-process theory (Wason & Evans, 1974), later popularized as "System 1" and "System 2" thinking (Kahneman, 2012). A similar dichotomy exists within metacognition (Flavell, 1979) where there are both "automatic" and "deliberate" kinds of metacognition. Automatic metacognition involves the use of metacognitive

"cues" (Ackerman, 2019), which are heuristics that provide information about the quality of the cognitive action. Examples include Feeling of Rightness (FoR) (Thompson et al., 2011), which refers to an intuition of answer correctness, as well as Feeling of Knowing (FoK) (Reder & Ritter, 1992) and Expectation Violation (Anderson & Fincham, 2014).

These cues differ from deliberate metacognition, which facilitates the communication of cognitive states (Shea et al., 2014), seeking help (Undorf et al., 2021), and regulating time investment (Ackerman, 2014; Undorf & Ackerman, 2017; Toplak et al., 2014). Recent AI research has already begun adopting these concepts (Bergamaschi Ganapini et al., 2025; Botvinick et al., 2019; Madan et al., 2021). Central to this is the dichotomy between metacognitive monitoring, which assesses object-level tasks, and metacognitive control, which allocates cognitive resources (Ackerman & Thompson, 2017). These processes work in tandem (Thompson, 2009; Ackerman, 2014; Shea et al., 2014), often triggered by metacognitive cues. Monitoring may also involve deliberate System 2 processes to transform cues into communicable representations (Shea et al., 2014). Once performance is assessed, metacognitive control enables higher-level reasoning, functioning as a "System 2 Intervention" for perceptual tasks (Thompson, 2009) or an evolution of solution strategies for reasoning tasks (Ackerman, 2014).

**The Case for Metacognitive AI.** We argue for adopting metacognition, involving self-monitoring and resource allocation inspired by cognitive science, as a ***general design principle in order to produce more efficient, safer, and more secure AI systems.*** Metacognition-based AI design would mean building up hierarchical structures of distributed collaborating or competitive agents and subsystems with dynamically reallocated resources. These metacognitive hierarchical knowledge and data systems would possess the following capabilities.

The implementation of metacognitive principles facilitates **greater efficiency** through adaptive resource deployment based on task priority, while enhancing **self-monitoring and anomaly detection** to recognize out-of-distribution (OOD) inputs, processing faults, and diverse anomalies caused by shifting datasets or malicious actors. This internal awareness further strengthens **robustness against adversarial attacks** by allowing models to introspect on vulnerabilities and reject suspicious inputs, alongside enabling **dynamic adaptation and recovery** from concept drift through autonomous model recalibration. Metacognition provides **better decision justification** by tracking reasoning quality, thereby improving auditability in critical domains. These capabilities address fundamental limitations in traditional statistical learning theory, which often prioritizes accuracy over complexity and feasible approximations (Horvitz, 1987; Russell & Wefald, 1991; Gershman

et al., 2015; Owhadi & Scovel, 2017). Just as humans switch strategies based on a balance of accuracy and computational cost (Thompson, 2009; Ackerman, 2014; Ackerman & Thompson, 2017; Kool & Botvinick, 2018; Lieder et al., 2026), deeper collaboration between cognitive and computer scientists can yield more resilient systems and a more profound understanding of intelligence. In sec. 6, we define research tasks that we believe are essential for navigating the next frontier of metacognitive AI research.

## 3. Alternative Views

One counter-argument to our position is that compute will continue to scale and resources will continue to get cheaper, ultimately diminishing any efforts towards more clever handling of limited resources (Sutton, 2019). This could be dubbed the "scale is all you need" stance. We would reply to this argument in two ways. First, we highlight that metacognition can be about more than just resource austerity. Meta-reasoning about uncertainty should lead to improved robustness at any scale. Second, smarter resource management has positive impacts on the economics, accessibility, and environmental impacts of AI. Successful AI systems of the future will improve on current systems along multiple objectives, not just task performance.

A second counter-argument is that metacognition is obviated by post-hoc correction, reasoning, or "generate-and-verify" pipelines (Cobbe et al., 2021), which select the best of multiple outputs to avoid the complexity of prospective decision-making. While these capacities improve AI capabilities (Wei et al., 2022; Kaplan et al., 2020), they exacerbate efficiency problems and are often infeasible in real-time and safety-critical domains. Metacognition manages these constraints by acting as a control layer to direct computational power ahead of time and only where strictly necessary (Russell & Wefald, 1991; Gershman et al., 2015; Kadavath et al., 2022).

A third counter-argument is that modern AI already has metacognition-like resource monitoring by different names or at other levels of abstraction. Indeed, our position echoes other recent calls for more metacognitive capacities in AI (Johnson et al., 2026), and there is already a push among recent work seeking more efficient AI systems through test-time adaptability (Rahmath P et al., 2025). Further, one could argue that resource efficiency has been moved from inference time to training time and deployment considerations such as selecting whether to deploy a small-but-inaccurate or larger-but-more-accurate model for a particular domain. While we agree that all of these are promising trends, the lack of unified theoretical foundations remains a barrier and progress is being made on isolated problems with bespoke methods. Our position is that ML and AI will benefit in the near-term from concerted efforts towards solving open problems in metareasoning and creating standardized frameworks for implementing and evaluating metacognitive systems.

A sharper version of this critique points to a number of existing techniques that already exhibit some metacognitive flavor: early stopping in training, early-exit networks (Rahmath P et al., 2025; Laskaridis et al., 2020), mixture-of-experts routing, cost-aware AutoML, and runtime model selection (Taylor et al., 2018; Howard et al., 2017). We view these as kindred efforts, but three properties distinguish them from what we mean by a metacognitive controller. First, *explicit separation*: the controller is a distinct supervisory layer with its own monitoring state and control signal, rather than a single knob folded into the object-level model. Second, *learned rather than tuned*: its parameters are optimized jointly with the object-level system in the framework introduced below (Eq. 1), instead of being settled by offline hyperparameter search and frozen at deployment. Third, *active at runtime*: the controller exercises its policy during the execution, so a system can adapt to the current environment and end-user requirements relying on a cheap mode for routine conditions and invoke an expensive one only when needed (e.g., when malicious attacks are detected), rather than committing to a single regime ahead of time. We see real value in this: it turns metacognitive *calibration*, *failure*, and *generalization* into first-class research objects (sec. 4), which would otherwise remain implicit within each of the methods above.

## 4. Challenges to Metacognitive AI and Responses

**Research Challenge 4.1.** *Developing metacognitive cues and controllers in a low-overhead manner in terms of computational cost and system complexity.*

The paradox of resource allocation is that it takes additional overhead to monitor and control resource usage. Yet, introducing a controller to budget resources can dramatically increase the overall performance of a system, even accounting for the added overhead of the controller itself. In such a metacognitive system, the sources of computational cost would come from the controllers and the cues. "Resource rationality" provides a framework for systems design while ensuring that overhead costs are managed (Russell & Wefald, 1991; Lieder & Griffiths, 2020; Griffiths et al., 2015; Lieder et al., 2026). To make this precise, let $f_{\mathbf{w}}(\mathbf{x}, \mathbf{c})$ be an object-level function or a family of functions with learnable parameters $\mathbf{w}$ that processes instances $\mathbf{x}$ and whose high-level behavior (and resource use) can be controlled via $\mathbf{c}$. The job of a meta-level policy $\pi$, which could have its own learnable parameters $\mathbf{w}'$, is to efficiently select $\mathbf{c}$ based on self-monitoring state $\mathbf{s}$ and/or on the data itself. Thus, $\pi_{\mathbf{w}'}(\mathbf{c}|\mathbf{x}_i, \mathbf{s})$ expresses a metacognitive policy for select-

ing a mode ($\mathbf{c}$) of handling a particular instance ($\mathbf{x}$) taking into account available resources ($\mathbf{s}$). The general idea of the resource-rational approach is to minimize a combined objective,

$$
\min_{\mathbf{w},\mathbf{w}'} \frac{1}{N} \sum_{i=1}^{N} \mathbb{E}_{\mathbf{c}_i \sim \pi_{\mathbf{w}'}} \left[ \text{loss}(\mathbf{x}_i, f_{\mathbf{w}}(\mathbf{x}_i; \mathbf{c}_i)) + \alpha \text{cost}(\mathbf{c}_i) \right] + \beta \|\pi_{\mathbf{w}'}\|, \tag{1}
$$

where $\|\pi_{\mathbf{w}'}\|$ represents the complexity or overhead of the metacognitive controller itself. The term $\text{cost}(\mathbf{c})$, on the other hand, expresses the cost of running a particular mode of $f$. For example if $f$ consists of a collection of models of varying complexity and performance, then $\mathbf{c}$ might index the different models and the job of $\pi$ is to select a model for a particular instance based on some efficient heuristic (Taylor et al., 2018). The $\text{cost}(\mathbf{c})$ term would then be the cost of running that particular model. Optimizing (1) given hyper-parameters $\alpha$ and $\beta$ results in a system that balances performance on the task and resource use.

The alternative to the metacognitive approach is to omit the controller $\pi$ and instead select a single model (a single $\mathbf{c}$) optimized for the amortized cost. Note that this is equivalent to the metacognitive objective (1) when the metacognitive policy is to always use the same mode $\mathbf{c}$ regardless of the instance or state, and where the overhead cost of this fixed policy is zero. From this perspective of resource-rationality, the perhaps surprising claim is that optimizing (1) can yield minima with lower overall loss despite the additional overhead of the metacognitive controller (Lieder et al., 2026). The controller "pays for itself" in terms of lower costs on particular instances. Precisely characterizing cases where the accounting works out in favor of (1) is embedded in the Research Tasks outlined in sec. 6: it requires better understanding the inherent trade-offs between task loss and resource cost (Task 6.1), and it requires developing efficient and robust cues or triggers for $\pi$ to act on (Task 6.2) such that the overhead $\|\pi_{\mathbf{w}'}\|$ is low.

While equation (1) sketches a formal framework for metacognitive strategies for inference, an analogous problem is faced in managing resources during the learning process itself. A resource-constrained learner may need to judiciously select its next training data – a process known as "active learning" (Settles, 2009). Similarly, decentralized training requires an aggregation strategy that makes secure and statistically efficient use of decentralized data (McMahan et al., 2017). In the case of learning, the investment of resources into a metacognitive controller is even more clearly beneficial because the overhead costs of a controller diminish when amortized over a model's lifetime.

**Research Challenge 4.2.** *Translating cognitive science concepts of metacognitive cues and controllers into precisely defined computational infrastructure.*

There are several groups that may help with this translation. Among them are the neurosymbolic AI community (Shakarian et al., 2023) and the cognitive modeling community known for platforms such as ACT-R and SOAR (Laird et al., 2017), which recently turned their attention to issues relating to metacognition (Lebiere et al., 2025). Another community includes those involved in hyper dimensional computing (Kanerva, 2009) where recent results have shown some metacognitive ability to reason about the use of different models for perceptual tasks (Sutor et al., 2022). For example, abductive learning (ABL) (Dai et al., 2019) was shown to be used with what can be viewed as a metacognitive cue in "ABL with new concepts" (*ABL_nc*) (Huang et al., 2023) where an OOD detector (effectively a metacognitve cue) is used to trigger reasoning about an unseen class – thereby improving performance. Along these lines, another neurosymbolic paradigm called NASR (Cornelio et al., 2023) uses a model trained to learn errors from an ML model that can lead to errors in reasoning.

However, we note that systems like *ABL_nc* and NASR both rely on model to cue correction – but this cueing model is itself susceptible to errors. This is also true of metacognition in humans, which is also known to fail (Fleming & Lau, 2014; Huda et al., 2018). This leads to our next research challenge.

**Research Challenge 4.3.** *Characterizing failures of metacognition.*

To address this challenge, it will be important to study the relationship between metacognitive performance and overall system performance. For the former, research in the cognitive science literature (Fleming & Lau, 2014) has provided some guidelines focused on metacognitive distribution of subject-assigned confidence scores. In that work, it is noted that the effectiveness of confidence scores relates to two factors: sensitivity (the separation between confidence assignment for correct and incorrect responses) and bias (the alignment of higher confidence scores with correct responses). If not decomposed, sources of metacognitive error may become conflated. Separately, recent work on artificial metacognition looked to establish a theoretical basis on the relationship in series of rigorous analytical results (Shakarian et al., 2025). For example, accurate metacognitive detection of errors in a multi-class classification setting may lead to poorer overall performance if class labels cannot be reassigned in a manner to improve precision of the reassigned classes. Such limits have implications for metacognitive control.

A separate class of failures arises because a metacognitive controller can make mistakes in ways that an object-level system cannot. The controller may misjudge its own com-

petence, i.e., treating its own confidence as reliable when it is not, or select an intervention that is locally rational but globally counterproductive, such as switching to an expensive aggregation rule precisely when the resource budget is most strained. These compounding meta-level errors are qualitatively different from object-level mistakes: they can destabilize the underlying learning process rather than producing a single incorrect output, and they therefore call for their own characterizations, benchmarks, and mitigations.

# 5. Metacognitive AI Transition to Practice: Case Study and Tool for Federated Learning

## 5.1. Federated Learning and its Vulnerabilities

Federated Learning (FL) is a decentralized ML paradigm that enables multiple clients to collaboratively train a shared model without exposing their private data (McMahan et al., 2017), becoming indispensable in privacy-sensitive domains like healthcare and finance. However, FL systems face an inherent trade-off between security and learning efficiency (Zhang et al., 2023; Yan et al., 2023), as introducing strong security mechanisms imposes overhead that can slow convergence. Transitioning to production is particularly challenging because conventional FL remains vulnerable to malicious attacks, such as data and model poisoning, that are accessible to adversaries with minimal expertise (Shejwalkar et al., 2022; Chuprov et al., 2025a). While defense mechanisms relying on distance metrics have been proposed to detect and exclude anomalous updates, these methods often introduce computational overhead and may inadvertently exclude benign clients, thereby reducing model quality and training efficiency. Importantly, the decentralized and iterative nature of FL implies that the system inherently possesses metacognitive features (Fig. 1), which, if properly leveraged, can resolve these persistent vulnerabilities (Shakarian, 2026).

## 5.2. Metacognition Capabilities Possessed by Federated Learning

The iterative structure of FL inherently mirrors the duality of metacognitive processes, separating the "object-level" task of learning from data (performed locally by clients) from the "meta-level" task of integrating and evaluating that learning (performed by the central server). This separation allows for the explicit realization of metacognitive functions that regulate the learning process as illustrated in Fig. 1.

**Metacognitive Monitoring** in FL may be realized through the systematic evaluation of client contributions before they are integrated into the global knowledge base. Instead of treating all updates as valid, the server employs monitoring functions ($\mathcal{M}$) to inspect statistical properties, such as the distance of an update from the global consensus or its

*Table 1.* From cognition theory to AI practice: overview of problems and potential practical solutions

| PROBLEM | SOLUTION | FL IMPLEMENTATION |
|---|---|---|
| **INFERENCE** | | |
| Fast and energy-efficient inference | Model selection (Taylor et al., 2018); early exit (Rahmath P et al., 2025) | Improved efficiency by selective aggregation and excluding untrustworthy clients (sec. 5.7) |
| Partially-observed decision process | Explore/exploit (Sutton & Barto, 2018); active inference (Friston et al., 2015) | Server infers client reliability via selective aggregation ($\mathcal{M}$) despite partial observability (sec. 5.2) |
| **LEARNING** | | |
| Learn from limited samples | Active learning (Settles, 2009) | Trustworthy client selection ensures model learns from reliable samples (sec. 5.6) |
| Aggregating decentralized learners | FL aggregation (e.g., FedAvg) (McMahan et al., 2017) | Adjustable aggregation with anomaly detection and trustworthiness evaluation (sec. 5.6) |

alignment with historical performance. Such systems invest meta-level resources to track "quality of learning" continuously, identifying updates that are anomalous, adversarial, or low-quality.

**Metacognitive Control** follows monitoring and refers to the active regulation of the aggregation process. Based on the assessment from the monitoring phase, the system exerts control by dynamically accepting, rejecting, or re-weighting client updates. This capability transforms aggregation from a static mathematical average into a dynamic decision-making process where the system autonomously decides what information is worth retaining.

**Self-Adaptation** may be achieved when the system modifies its own strategies in response to environmental feedback. For instance, if the monitoring function detects a spike in data heterogeneity or a coordinated attack, the system can adapt its hyperparameters (such as the strictness of exclusion thresholds or the learning rate) to maintain stability.

Together, these features significantly enhance the system's intelligence level and its deployment capabilities (see Table 1). **Learning efficiency** is improved because the model avoids "unlearning" progress caused by detrimental updates, leading to faster convergence. Better **performance** is achieved through primarily selecting high-utility updates for the aggregation. **Robustness** is strengthened because the system can autonomously identify and neutralize adversarial inputs that would otherwise corrupt the learning trajectory.

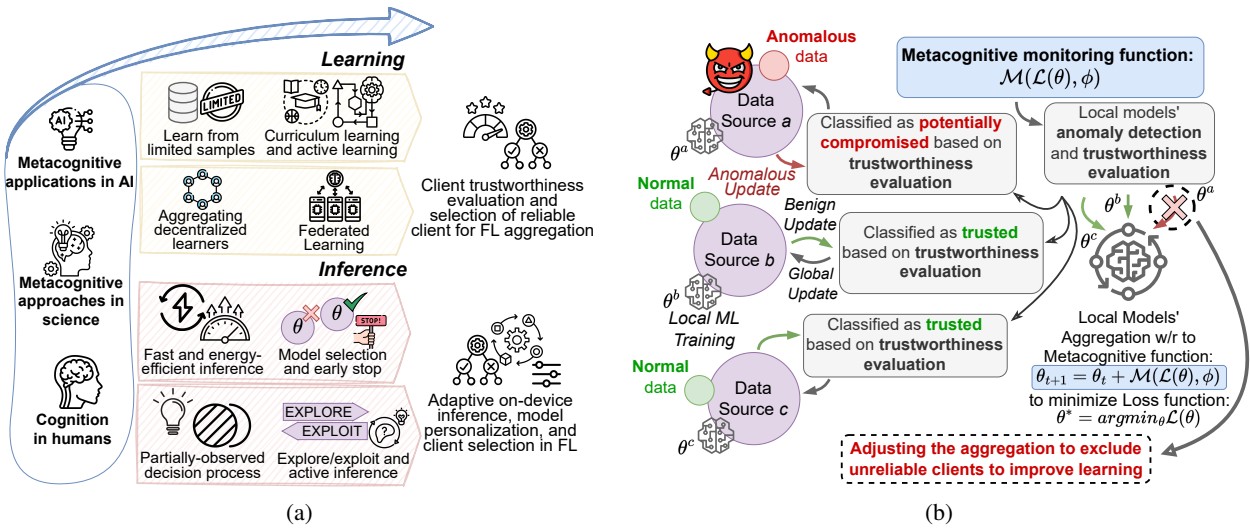

*Figure 1.* Metacognitive approaches in AI for enhanced learning and inference: (a) overview of problems in learning and inference, potential metacognitive solutions, and their realization in the FL case; (b) mechanism of a metacognitive monitoring function ($\mathcal{M}$) within an FL system, illustrating how client trustworthiness evaluation and selective aggregation are used to filter unreliable updates and improve the learning of the global model

### 5.3. InteFL Framework

To translate metacognitive principles into practice and exploit these latent system characteristics, we introduced *InteFL* (Korobeinikov et al., 2026), a framework for designing and evaluating metacognition-enabled FL applications. InteFL employs a two-layered architecture (Fig. 2) to benchmark robustness. At the framework level, an AI agent enables natural language configuration and post-experiment optimization through a feedback loop. System-wide, the framework enables the treatment of FL characteristics as metacognitive features, conceptualizing learning through planning (updates), monitoring (quality), and evaluating (aggregation). This is formalized by augmenting the primary loss $\mathcal{L}(\theta)$ with a monitoring function $\mathcal{M}(\mathcal{L}(\theta), \phi)$, resulting in the update rule $\theta_{t+1} = \theta_t + \mathcal{M}(\mathcal{L}(\theta), \phi)$, where $\phi$ denotes indicators such as distance-based trust metrics (Blanchard et al., 2017; Mhamdi et al., 2018; Pillutla et al., 2022; Zatsarenko et al., 2026a).

Below, we demonstrate how our framework can be employed to investigate and optimize the FL parameters for user-specified use-cases. The process of the AI-assisted case-study generation and refinement, along with the AI agent's reasoning on collected metrics, is fully detailed in (Korobeinikov et al., 2026).

### 5.4. Translating Metacognitive Theory into Federated Learning Practice

We now make the connection between InteFL and the resource-rational objective (1) precise. At round $t$, let $\mathbf{x}_t$ stand for the set of local client updates $\{\theta_t^{(k)}\}_{k=1}^{K}$ received by the server, and let the object-level function $f(\mathbf{x}_t; \mathbf{c}_t)$ realize one aggregation step, $\theta_{t+1} = A_{\mathbf{c}_t}\left(\{\theta_t^{(k)}\}_{k=1}^{K}\right)$. The mode $\mathbf{c}_t \in \mathcal{C}$ indexes the available aggregation strategies (for instance, FedAvg (McMahan et al., 2017), Multi-Krum (Blanchard et al., 2017), a PID-based exclusion rule (Zatsarenko et al., 2026b; 2025), or a trusted-subset projection), and crucially different strategies $\mathbf{c}_t$ may be selected at different steps $t$. A self-monitoring state $\mathbf{s}$ aggregates round-by-round signals such as per-client trust scores, gradient-distance statistics, and exclusion history. The metacognitive policy $\pi_{\mathbf{w}'}(\mathbf{c}_t \mid \mathbf{x}_t, \mathbf{s})$ is realized by InteFL's adjustable aggregation mechanism: its learnable parameters $\mathbf{w}'$ are the exclusion thresholds and other rule-specific hyperparameters that the framework tunes against observed performance. The object-level term $\text{loss}(\mathbf{x}_t, f(\mathbf{x}_t; \mathbf{c}_t))$ in (1) is the validation error of the global model after aggregation; the term $\alpha \, \text{cost}(\mathbf{c}_t)$ captures the per-round compute and communication price of mode $\mathbf{c}_t$, which varies across strategies: robust rules that involve pairwise client comparisons are considerably more expensive than simple averaging-based ones. Finally, $\beta \|\pi_{\mathbf{w}'}\|$ accounts for the controller's own overhead, namely the anomaly detector and its threshold-update loop. Because this overhead is paid at training time and amortized over the model's deployment lifetime, the FL setting is especially favorable for metacognitive control: an expensive meta-decision at learning time is far more tolerable than equivalent inference-time overhead.

The same realization clarifies when this controller should and should not pay off. In settings with substantial client heterogeneity, distribution drift, or adversarial presence, $\pi_{\mathbf{w}'}$ actively switches the aggregation mode round by round, and

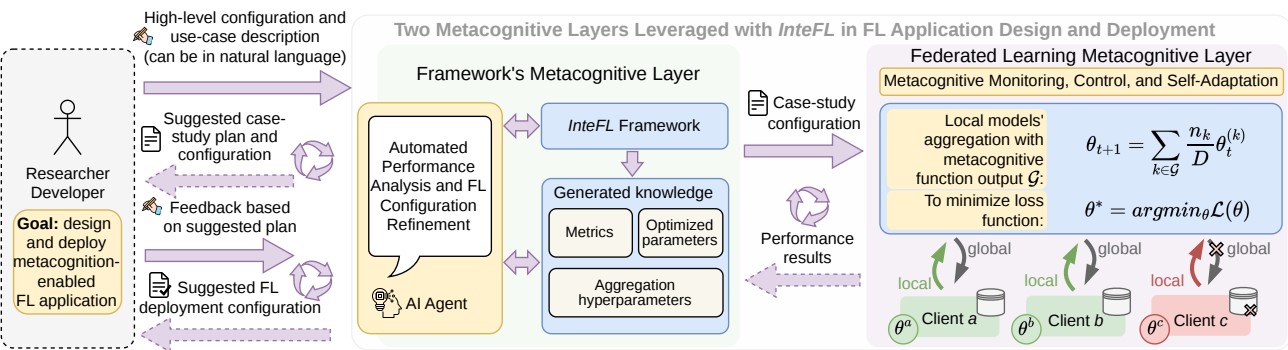

*Figure 2.* The two-layered architecture leveraged by *InteFL*. The framework bridges high-level user intent and practical deployment by combining an AI-assisted design layer with a FL metacognitive layer for applications design and deployment

the gain from instance-specific adaptation outweighs the controller's overhead. In benign, stationary settings, the optimal policy degenerates to selecting the cheapest mode (effectively FedAvg) on every round, so the system collapses to its non-metacognitive baseline. This is precisely the property a resource-rational controller is supposed to have, and it distinguishes the construction from offline algorithm selection or static multi-objective tuning, where a single configuration is fixed before deployment and is not re-evaluated against the running self-monitoring state $\mathbf{s}$.

The mapping is also independent of the underlying object-level model. Although the case study below trains a CNN classifier, the same approach is already available in InteFL for LLM training pipelines (BiomedBERT and GPT-2 on MedQuad and NER datasets), supporting our broader position that metacognitive design is architecture-agnostic.

### 5.5. Case Study: Medical Image Classification

Within the case study, the FL setup was assessed against two objectives: (1) the selection of an aggregation algorithm satisfying characteristics such as the fastest convergence of the aggregated model, the highest performance in detecting and removing anomalous models, and the least computational complexity; and (2) for the selected algorithm, the benchmarking of how the optimization of its parameters may enhance FL metacognitive capabilities.

FL was evaluated on the OctMNIST optical coherence tomography retinal imaging dataset using a CNN architecture for the classification task. The system was executed with 20 aggregation rounds and 20 participating clients, 4 of which flipped 100% of labels on their local training subsets, introducing label-level noise to the model.

Various aggregation algorithms perform filtering of anomalous model updates differently. Some algorithms are able to immediately react and exclude such models from aggregation; others are unable to exclude all such models due to limitations imposed by the algorithm's design and required

parameters. These factors result in the delayed convergence of an aggregated model. To address the first objective, six aggregation algorithms representing different metacognitive approaches were benchmarked, including control-based adaptive filtering and distance-based methods. Hyperparameters for every algorithm were set according to theoretical recommendations to allow fair comparison under theoretically perfected conditions. Once the optimal algorithm was chosen, executions with varied hyperparameters were conducted to investigate improvements to the algorithm design. Detailed experimental configurations and results are available in (Korobeinikov et al., 2026).

### 5.6. Addressing Challenges in Aggregating Decentralized Learners

FL executed with Trimmed Mean (Yin et al., 2018; Blanchard et al., 2017) and Bulyan (Mhamdi et al., 2018) aggregations resulted in the inability of the centralized model to converge to the same loss values as with other aggregation algorithms. As detailed in (Korobeinikov et al., 2026), these results align with client exclusion performance metrics. Trimmed Mean and Bulyan yielded lower exclusion accuracy and precision compared to other methods, failing to exclude all anomalous clients at certain aggregation rounds. This results in the higher loss values of the aggregated model. These results suggest that some algorithms will render better metacognitive capabilities in real-world FL setups, as they are able to better react to environmental conditions resulting in anomalous client-supplied models.

### 5.7. Optimizing Learning Efficiency via Hyperparameter Tuning

We continued investigation of optimal deployment hyperparameters with the PID-based algorithm (Zatsarenko et al., 2026b; 2025), as it yielded the fastest convergence among all algorithms. Optimization of robust aggregation algorithm hyperparameters to the expected data heterogeneity pattern greatly influences both the speed of convergence and

the ability of FL aggregation algorithms to adequately react to substandard models. Combining observations on loss and model exclusion performance described in (Korobeinikov et al., 2026), PID aggregation with the hyperparameter characterizing the number of standard deviations for client exclusion set to 2.5 results in the best convergence and the most accurate malicious client exclusion. This indicates that this algorithm with the selected set of hyperparameters allows FL to implement its metacognitive capabilities most effectively.

This case study demonstrates how our framework helps to leverage and test metacognition capabilities in the design of FL applications. By simulating specific execution conditions, we first identified the aggregation algorithm yielding the fastest convergence and then further optimized its hyperparameters. This iterative process confirms that treating FL characteristics as metacognitive features enables the efficient design of robust, domain-specific applications tailored to complex data heterogeneity.

### 5.8. Addressing Emerging Metacognitive Challenges with *InteFL*

**Designing Applications with Metacognitive Capabilities.** Developing metacognitive AI systems with self-monitoring and dynamic control is vital for efficient applications (Shakarian & Wei, 2025; Chuprov et al., 2025b), yet systematic benchmarking remains challenging. *InteFL* bridges this gap by enabling the validation of capabilities like resource optimization through robust aggregation, allowing users to prioritize valuable updates and prevent resource waste on untrustworthy clients. The framework facilitates testing self-monitoring by leveraging the natural metacognitive features of FL involving planning, monitoring, and evaluating. By supporting the recreation of anomalies like OOD data or poisoning, it allows researchers to benchmark strategies like Multi-Krum (Blanchard et al., 2017) and generates metrics to quantitatively measure the system's self-monitoring ability and impact on performance.

**Robustness Against Adversarial Attacks and Execution Environment Factors.** Greater robustness against adversarial attacks is achieved when metacognition helps to reflect on vulnerabilities and respond adaptively. Our framework provides a testbed for this by enabling the configuration and execution of adversarial attacks. In our practical case (Korobeinikov et al., 2026), this is handled by adjustable aggregation mechanisms acting as a metacognitive control. It uses a monitoring function for model anomaly detection to assess client contributions, allowing for benchmarking strategies that serve as an adaptive response to maintain performance under attacks. This applicability extends to data heterogeneity where clients hold local and variable-quality data, a capability systematically testable with *InteFL*. The

framework yields performance metrics such as loss and accuracy to quantify how effectively different strategies adapt to attacks or abnormal conditions.

**Better Decision Justification and Explainability.** Explainability is enhanced by metacognition that supports trust and accountability. From the server's perspective, FL is a partially-observed decision process lacking full insight into a client's local data. Metacognitive monitoring tackles this directly by allowing the server to infer the reliability and quality of client contributions even with incomplete information. Our framework allows researchers to test this capability by creating a comprehensive record for each experiment where the centralized configuration file documents the exact initial setup, parameters, and intent. By automatically saving and visualizing metrics, the framework links the experimental design to the resulting system behavior, rendering the decision-making process more transparent.

### 5.9. Addressing Open Research Questions in Metacognitive AI

Beyond testing specific capabilities, the *InteFL* design provides a practical methodology for addressing several open research questions in metacognitive AI (Shakarian & Wei, 2025).

A key challenge is *designing metacognitive AI that adapts to rapidly changing and unpredictable environments. InteFL* addresses this by allowing the simulation of such conditions for testing, using scheduling and configuration features to evaluate how strategies generalize to unforeseen changes like dynamic attacks.

Another significant hurdle is *enabling AI systems to autonomously modify learning strategies for continuous self-improvement.* Our framework provides an evaluation environment for such algorithms, allowing researchers to test strategies with self-modification capabilities and validate whether autonomous modifications successfully improve performance using generated metrics.

Making *metacognitive self-assessment interpretable* is crucial for ensuring trust. *InteFL* facilitates this by linking experimental intent to system behavior, enabling researchers to correlate setup with round-by-round results to understand specific decisions like model exclusion.

The *difficulty of developing benchmarks* is addressed by providing an AI-assisted generator for complex and reproducible scenarios. Centralized configuration and runtime poisoning allow researchers to systematically benchmark metacognition-enabled applications against consistent baselines using standard or custom datasets.

## 6. Call to Action

Papers proposing metacognitive frameworks are becoming more prevalent, as seen in recent summaries of the field (Shakarian, 2026; Johnson et al., 2026). However, current work has largely focused on metacognitive monitoring, specifically detecting errors in ML outputs, while the downstream use of these cues remains less explored. Notable exceptions include the use of metacognitive cues to ensemble models in OOD settings (Leiva et al., 2026) and regulating expensive LLM usage (Yang et al., 2025). To fully realize the advantages of these systems, the cognitive psychology concept of metacognitive control (Ackerman & Thompson, 2017) must be instantiated computationally. Research must establish rigorous theoretical foundations that link detection performance to control tasks such as correction, security, and resource regulation. This should lead to performance benchmarks that evaluate downstream control functions rather than just detection accuracy (Ngu et al., 2025).

**Research Task 6.1.** *Rigorously characterize tradeoffs between task performance and computational resources for learning and inference tasks.*

Balancing resource costs with task performance is a multi-objective optimization problem where a metacognitive architecture must support more than one solution. A minimal system might support binary switching between expensive, accurate modes and cheap, approximate modes. While intuitive, characterizing the tradeoffs between time, memory, energy, and accuracy remains an incompletely solved problem (Hay, 2016; Owhadi & Scovel, 2017). Although work exists on PAC learnability (Valiant, 1984), anytime algorithms (Zilberstein, 1996), and energy-efficient models (Howard et al., 2017), there is a lack of a comprehensive framework for task and resource tradeoffs that would support general metacognitive function. Recent applications to LLMs show that metacognitive triggers can allow for the use of smaller models while reserving larger models for correction (Shamim, 2025; Yang et al., 2025).

**Research Task 6.2.** *Develop robust and efficient metacognitive cues.*

A metacognitive system requires cues or triggers to provide reliable information to the meta-level system. Existing work includes Error Detection Rules (Kricheli et al., 2024; Xi et al., 2025; Lee et al., 2024) which increase precision under certain distributional assumptions (Shakarian et al., 2025). Other strategies allow neural networks to exit early or iterate through recurrence to monitor uncertainty (Laskaridis et al., 2020; Spoerer et al., 2020; Hartmann, 2018). Proper cues should go beyond error detection to support rich control policies where an agent understands the underlying causes of its mistakes. Mechanisms from Explainable AI, such as

LIME (Ribeiro et al., 2016) or SHAP (Lundberg & Lee, 2017), can be leveraged internally to identify influential features in erroneous decisions. Advanced agents might use credit assignment mechanisms (Barto et al., 1983) to reflect on failure modes. These cues must be produced in near-constant time to avoid substantial reasoning overhead (Russell & Wefald, 1991).

**Research Task 6.3** *Develop well-calibrated metacognitive controllers and integrate with uncertainty quantification.*

Existing work includes using a second model to predict the failures of a primary model (Daftry et al., 2016; Ramanagopal et al., 2018). These cues function as probabilities of success, but they must be well-calibrated to be reliable (Dawid, 1982). Methods such as temperature scaling (Guo et al., 2017), conformal prediction (Park et al., 2024), and abductive inference (Leiva et al., 2026) should be adapted to the meta-level (Ovadia et al., 2019; Yang et al., 2024). In sequential decision-making, uncertainty regarding the state of the world or the best action requires a metacognitive policy to decide when to stop gathering information and act (Russell & Wefald, 1991; Hay et al., 2014; Friston et al., 2015; Callaway et al., 2022). This creates a unified sense where rewards and certainty are both resources managed through meta-actions (Lin et al., 2015; Hay, 2016; Bénon et al., 2024; Lieder et al., 2026).

## 7. Conclusion

In this paper we argue that various communities, from cognitive scientists to AI/ML engineers, should join their efforts of further developing the metacognition concepts and transitioning them into AI design practice, making them the centerpiece of future AI systems. We note that over decades of research in cognitive science, metacognition has been posited as a key enabler of general intelligence (Lenat et al., 1983; Johnson et al., 2026). However, we take a more pragmatic view of artificial metacognition considering it as a paradigm with dealing with the growing computational cost of AI systems and providing the practical way of implementing it in real life AI systems.

## Acknowledgements

Leon Reznik, Raman Zatsarenko, Dmitrii Korobeinikov were supported in part by the US National Science Foundation (award 2321652).

Paulo Shakarian was supported by ARO grant W911NF-24-1-0007.

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
