# Title Suppressed for Anonymity

Anonymous Authors, *Suppressed for Anonymity*

*Abstract*—We argue that metacognition should be the guiding principle for designing and deployment of Federated Learning (FL) applications. The metacognitive features in FL, such as monitoring and control, allow for the dynamic adaptation to the environment, client updates, enhanced user interaction, and optimized resource-aware learning. We introduce IntelliFL, the first framework that facilitates designing metacognitive FL systems by providing mechanisms and tools to investigate how possible changes in execution and learning conditions impact FL performance and security, and then to choose or optimize the learning structure and process based on their evaluation. We demonstrate the framework application in practice on the cases showing how the choice of FL hyperparameters such as model aggregation algorithms can be dynamically adapted or fine-tuned in response to changing execution environment, including data quality variations, network conditions, or adversarial attacks.

Federated Learning (FL) has attracted significant attention since its inception in 2016 [1]. It iteratively trains local models across multiple data source devices and then combines them into a single global model using an aggregation algorithm. As the demand for scalable computation and distributed model training continues to grow, FL has become increasingly valuable, offering a prominent solution to meet both computational and security requirements of contemporary AI systems. Current research predominantly investigates FL methodological foundations, treating FL as a decentralized optimization problem [2], [3], [4], but offers very limited guidance and tools on how FL systems should be designed, configured and deployed in real-world settings. In practice, the decentralized nature of FL necessitates consideration of operational conditions inherent to distributed systems, where data and model quality degradation can be caused by both constant environment changes and possible adversarial attacks. Similar challenges have recently been identified in broader ML system deployments, where infrastructure-level perturbations and adversarial manipulations jointly impact end-model performance, motivating knowledge-integration–based system designs for attack detection and classification [5]. While conventional FL with aggregation strategies such as FedAvg lacks mechanisms to mitigate these threats, the number of already proposed dependability enhancement techniques complicates FL deployment into end-user applications, making it challenging to determine which strategy and its hyperparameters are appropriate for a given application and its execution environment.

Available FL tools (see Table 1) were designed with the major goal to assist in fast prototyping of existing FL algorithms, providing interfaces and APIs for designing FL execution. Yet, most remain focused on the task of conventional distributed ML model training and simulation of its inherent aspects rather than researching FL dependability and investigation of deployment parameters. Flower provides a backend for cross-platform experiments, FedML emphasizes scalability and Edge deployment, EasyFL reduces development overhead through modular APIs, and ByzFL explicitly targets Byzantine-robust aggregation benchmarking. The application of these tools is not trivial due to their limited flexibility in assessing FL robustness under conditions inherent to real-world systems that include data quality variation and adversarial attacks.

The challenges that come with existing tools demonstrate the need to significantly enhance the intelligence level in FL frameworks. In this paper, we argue and empirically demonstrate that understanding and application of FL as a *metacognitive* learning paradigm, consistent with the broader group of methods outlined in [11], can help to design more resilient real-world applications that would be able to withstand

2026 Suppressed for Anonymity
Suppressed for Anonymity

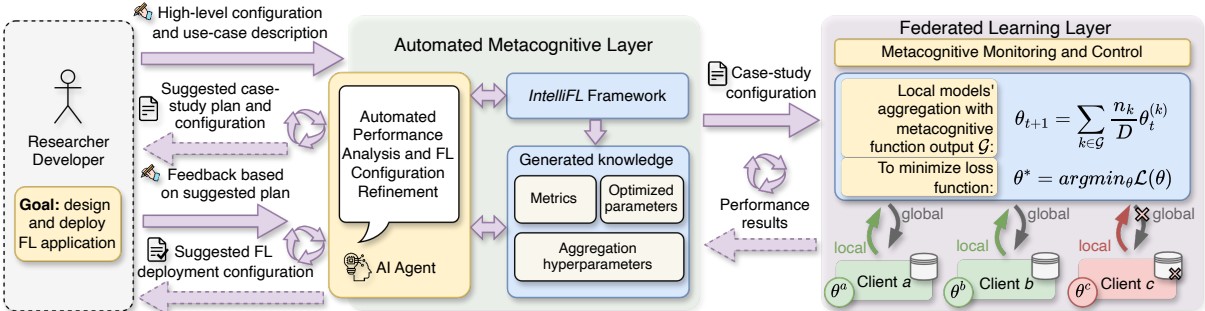

**FIGURE 1**: Integration of metacognitive objectives in the FL application design process. User-defined experiment configuration is assisted with AI agent and experiment result-driven feedback loop, while metacognitive monitoring and control are employed through FL adaptive aggregation.

**TABLE 1**: Comparison of *IntelliFL* capabilities with other FL frameworks and tools

| Framework | Capabilities[a] | | | | | | | | |
|---|---|---|---|---|---|---|---|---|---|
| | Dynamic attacks [b] | Centralized config | Robust agg. evaluation | Experiment scheduling | Config-based datasets | Metrics visualization | AI-agent case study | Optimal config. select. | Artifacts UI dash |
| Flower [6] | ✗ | ✗ | ✗ | ✓ | ✗ | ✗ | ✗ | ✗ | ✗ |
| FedML [7] | ✗ | ✗ | ✗ | ✓ | ✓ | ✓ | ✗ | ✗ | ✓ |
| EasyFL [8] | ✗ | ✓ | ✗ | ✓ | ✗ | ✓ | ✗ | ✗ | ✗ |
| FedEasy [9] | ✗ | ✓ | ✗ | ✓ | ✓ | ✓ | ✗ | ✗ | ✗ |
| ByzFL [10] | ✗ | ✓ | ✓ | ✓ | ✓ | ✓ | ✗ | ✗ | ✗ |
| ***IntelliFL*** | ✓ | ✓ | ✓ | ✓ | ✓ | ✓ | ✓ | ✓ | ✓ |

[a]Available out of the box.
[b]When the attack model and/or its intensity changes during FL execution.

challenges stemming from changing execution environment and heterogeneous data conditions.

To summarize the contributions, in this paper we: **(i)** present *IntelliFL*[1], a novel, configuration-driven framework for benchmarking, comparing, and verifying FL procedures and strategies; **(ii)** investigate the integration of metacognitive functions to enhance framework's operational intelligence and stabilize model convergence; **(iii)** prove theoretically and verify in practice that the incorporation of these metacognitive techniques, specifically self-monitoring and anomaly exclusion, results in faster learning convergence; **(iv)** demonstrate the framework's practical utility through the case study, showing how *IntelliFL* guides the design, performance benchmarking, and optimal hyperparameter configura-

tion of adaptive FL systems.

## INTELLIGENT FRAMEWORK FOR METACOGNITIVE FL

Metacognition refers to an intelligent system's capacity to monitor, evaluate, and regulate its own cognitive processes [12]. FL naturally embodies these characteristics, as its decentralized and iterative learning structure enables continuous assessment of client-side learning performance and adaptation of local models, while the aggregation process provides a mechanism for adjusting global training parameters. Selecting the models and methods for aggregation allows to optimize the overall learning process. By leveraging metacognitive processes such as monitoring and control, FL practitioners can enhance model adaptation, improve communication efficiency, and support more effective error correction, ultimately enabling the AI systems that are aware of and responsive to their environment as well as their own learning progress and needs. In this view, metacognition becomes the guiding design principle for practical FL deployments.

To facilitate implementation of metacognitive principles into FL, we introduce the *IntelliFL* Framework, a novel platform for designing, fine-tuning, and evaluating FL systems. Our tool enables to benchmark and iteratively refine FL configurations across diverse data modalities, quality characteristics, distributions, and aggregation algorithm settings. Metacognitive integration in our framework is twofold. On the one hand, the framework itself is metacognitive, as it incorporates the AI agent module that adapts to user requests, which is achieved with the feedback loop that allows users to iteratively refine experiment configurations through natural language conversation. After experiment execution, collected results are analyzed by an AI agent, and recommendations regarding fur-

[1]Suppressed for Anonymity

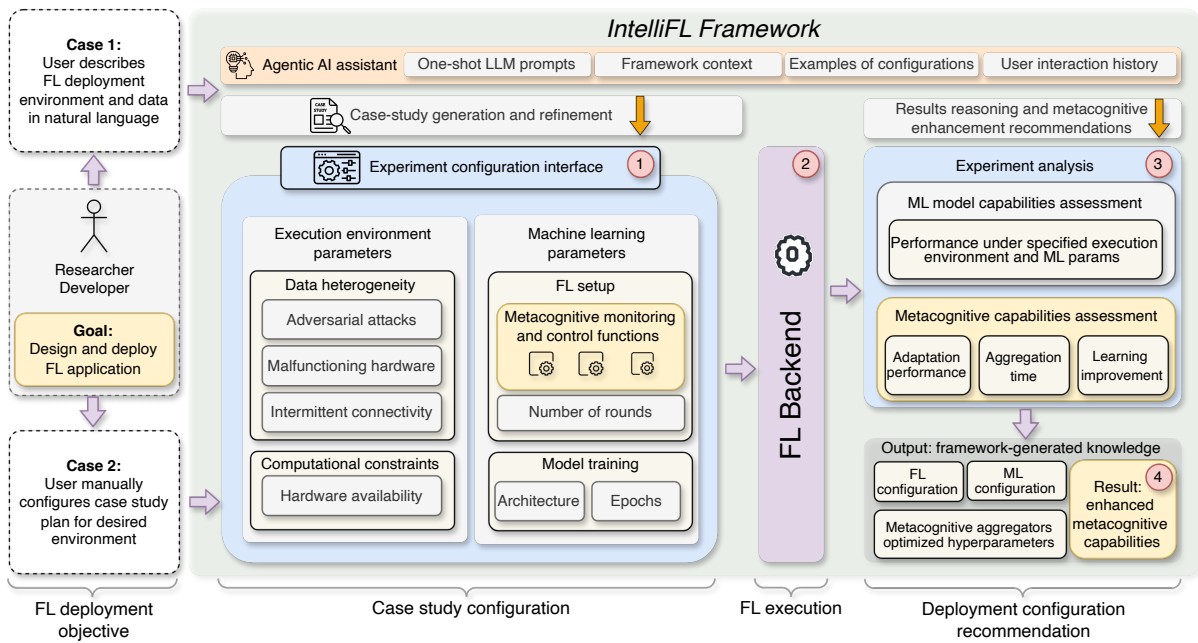

**FIGURE 2**: *IntelliFL* framework enables investigation of the relation between execution environment and machine learning (ML) parameters for the target domain. Based on the knowledge accumulated during systematic benchmarking, the framework provides the recommended FL deployment configuration that yields overall better metacognitive capabilities, with the robust aggregation algorithm hyper parameters tailored to specified execution environment conditions.

ther deployment configuration optimization and refinement are provided, which embodies post-experiment feedback loop. *IntelliFL* users may adopt the agent's suggestions or derive their own configurations for the target FL deployment based on their observations. On the other hand, *IntelliFL* implements metacognition within FL through the integration of adaptive robust aggregations which adjust based on the assessment of client updates, incorporating metacognitive monitoring and control during the experimental execution of FL. Figure 1 offers a high-level overview of *IntelliFL* integration with a metacognitive paradigm, illustrating the interaction between the base FL layer, the automated metacognitive layer, and the user-driven deployment configuration refinement process. *IntelliFL* provides a platform for the optimization of the FL design process with the help of increased intelligence and usability for FL practitioners.

## System Approach to FL Application Design

FL is a decentralized learning paradigm in which a set of participating clients $\mathcal{A} = \{1, ..., \mathcal{N}\}$ collaboratively trains a shared model. At round $t$, the server broadcasts $w_t \in \mathbb{R}^d$ to clients, who minimize their local objective $\mathcal{L}_t^{(i)} = \mathbb{E}_{(x,y) \sim \mathcal{D}_i}[\ell(f_w(x), y)]$ to produce updates

$w_t^{(i)} = w_{t-1}^{(i)} - \eta \nabla \mathcal{L}_t^{(i)}$. The server uses an *aggregation strategy* to update the global model; e.g., FedAvg computes $w_{t+1} = \sum_{i \in \mathcal{A}} \frac{|\mathcal{D}_i|}{\sum_{j \in \mathcal{A}} |\mathcal{D}_j|} w_t^{(i)}$. More sophisticated and robust strategies can exclude anomalous clients, reducing aggregation to $\mathcal{S} \subseteq \mathcal{A}$.

Optimizing FL settings for real world conditions is critical, including the selection of robust aggregation strategies and fine-tuning of their hyperparameters. Moreover, the algorithm selection is based on the target system constraints, e.g., outdated hardware. Metacognitive features, such as self-monitoring and control, are inherent to FL robust aggregation algorithm's execution. These methods dynamically assess updates affected by environmental factors or attacks. Subsequently, the server adjusts the learning strategy via this feedback mechanism. In the FL design stage, it is necessary to analyze the aggregation performance under conditions mimicking real-world systems. Consequently, algorithm selection and hyperparameter fine-tuning are influenced by the requirements, constraints, and anomalies anticipated in the target system.

Moreover, factors such as the data modalities, task type, and resources further influence the performance. We propose viewing FL as a parametric

system defined by (1) execution environment parameters (hardware, heterogeneity) and (2) ML parameters (architecture, robust aggregation). We argue that environment parameters should dictate ML parameters. Performance gains arise from enhanced metacognitive control where robust algorithms filter anomalous models. To facilitate investigating optimal parameters, we propose *IntelliFL*, a framework allowing one to achieve these objectives through systematic benchmarking.

## *IntelliFL* Application Workflow

Our framework serves as a configuration-driven platform for benchmarking, evaluating, and optimizing metacognitive FL applications. As illustrated in Figure 2, the workflow comprises four stages: (1) configuration, (2) execution via Flower as a backend [6], (3) analysis, and (4) knowledge extraction. In the initial phase (Figure 2, (1)), users define execution environment parameters, such as data heterogeneity, adversaries, and hardware constraints, and ML parameters, including robust aggregation strategies and hyperparameters. These settings can be selected based on theoretical recommendations (e.g., anticipated anomalies) and are strictly validated to prevent misconfiguration.

Once configured, *IntelliFL* uses Flower for distributed training (Figure 2, (2)), collecting standard metrics and aggregation data, such as anomalous client exclusion. These drive the analysis (Figure 2, (3)), where metacognitive objectives like dynamic control are assessed via exclusion indicators. Validating the filtering of substandard models enables knowledge extraction (Figure 2, (4)) to identify optimal ML settings and hyperparameters for the target environment.

## Agentic AI-Assisted Configuration and Development

Experienced users can manually specify execution and ML parameters while remaining safeguarded by strict validation mechanisms. Alternatively, we provide a natural-language interface that utilizes one-shot LLM prompting to automatically generate configurations, which we position as a unique feature that other research and benchmarking oriented FL frameworks lack. By incorporating context such as documentation, validation logic, and user history, the AI agent generates grounded case study designs. This integration embodies our framework's metacognitive aspect, allowing users to interactively refine plans and simplifying onboarding until they are familiar enough to opt for manual configuration. Following execution, analysis can be performed manually or with AI assistance. In the latter case, the agent performs a reasoning

step, utilizing collected metrics and the original request context to infer the optimal set of FL parameters tailored to the specific user requirements and execution conditions.

## ADDRESSING EMERGING METACOGNITIVE CHALLENGES WITH *IntelliFL*

### Designing Applications with Metacognitive Capabilities

Developing metacognitive AI systems with self-monitoring and dynamic control is vital for efficient applications [11], yet their systematic benchmarking and evaluation remains challenging. *IntelliFL* bridges this gap (Figure 2, left) by enabling the validation of capabilities like resource optimization through robust aggregation. This allows users to prioritize valuable updates, preventing resource waste on untrustworthy clients and avoiding model performance degradation. Furthermore, *IntelliFL* facilitates testing self-monitoring by leveraging FL's natural metacognitive features involving *planning* (distributing models), *monitoring* (inspecting updates), and *evaluating* (deciding which to aggregate). The framework supports research by re-creating anomalies like out-of-distribution data or poisoning, allowing researchers to benchmark strategies like Multi-Krum [13] or MADE-PI [14]. Finally, *IntelliFL* generates ML and aggregation metrics to quantitatively measure the system's self-monitoring ability and impact on performance.

### Robustness Against Adversarial Attacks and Execution Environment Factors

Greater robustness against adversarial attacks is achieved when metacognition helps to reflect on vulnerabilities and respond adaptively. Our framework provides a testbed for this by enabling the configuration and execution of adversarial attacks. In our practical case, this is handled by adjustable aggregation mechanisms (e.g., PI-based [14]) acting as a metacognitive control. It uses a *Model Anomaly Detection* monitoring function to assess client contributions, allowing for benchmarking strategies that serve as an "adaptive response" to maintain performance under attacks. This applicability extends to data heterogeneity, where clients hold local, variable-quality data, which is a capability systematically testable with *IntelliFL*. The framework yields performance metrics, such as loss and accuracy, to quantify how effectively different strategies adapt to attacks or abnormal conditions.

## Better Decision Justification (Explainability)

Explainability is enhanced by metacognition that supports trust and accountability. From the server's perspective, FL is a partially-observed decision process: it lacks full insight into a client's local data. Metacognitive monitoring directly tackles this by allowing the server to infer the reliability and quality of client contributions, even with incomplete information. Our framework allows to test this capability by creating a comprehensive record for each experiment. The centralized configuration file documents the exact initial setup, parameters, and intent of the experiment. The framework's automated output saves all metrics to files and visualizes certain metrics.

## Addressing Open Research Questions in Metacognitive AI

Beyond testing specific capabilities, *IntelliFL* design also provides a practical methodology for addressing several open research questions in metacognitive AI [11].

*Generalization to Diverse Dynamic Environments.* A key challenge is designing metacognitive AI that adapts to rapidly changing, unpredictable environments. *IntelliFL* addresses this by simulating such conditions for testing. Its scheduling and configuration features allow researchers to evaluate how metacognitive strategies (e.g., aggregation algorithms) generalize to unforeseen changes like dynamic attacks, enabling direct comparisons of their robustness.

*Designing for Continuous Self-Improvement.* Enabling AI systems to autonomously modify learning strategies for continuous improvement is a significant hurdle. Our framework provides an evaluation environment for such algorithms, allowing researchers to test strategies with self-modification capabilities (e.g., resource-based aggregation adjustments). By introducing anomalies during experiments, users can leverage generated metrics (like loss history) to validate whether the system's autonomous modifications successfully improve performance and recovery.

*Interpreting Metacognitive Processes.* Making metacognitive "self-assessment" interpretable is crucial for ensuring trust. *IntelliFL* facilitates this by linking experimental intent (configuration) to system behavior (output records). Researchers can correlate the setup with round-by-round results, such as loss history and client participation, to understand specific decisions, like model exclusion, thereby rendering the internal decision-making process transparent.

*Benchmarking Datasets and Baselines.* A major difficulty lies in developing benchmarks to evaluate metacognition. *IntelliFL* addresses this by providing a generator for complex, dynamic, and reproducible scenarios. It supports testing on standard datasets (e.g., FEMNIST, PneumoniaMNIST) or custom data. Through centralized configuration and runtime poisoning features, researchers can create challenging benchmarks to systematically evaluate applications with metacognitive capabilities.

## ACCELERATING CONVERGENCE IN FL WITH METACOGNITION

In this section, we provide theorems showing that metacognition functionalities that include self monitoring, anomaly detection and exclusion of anomalous actors, accelerate learning convergence. We formalize metacognition in FL as a higher-order process that observes, evaluates, and regulates the learning dynamics across clients and communication rounds, rather than directly optimizing the task loss. By endowing the system with the ability to reason about its own training state, such as detecting of adversarial updates, metacognition in FL allows to selectively modulate client participation and aggregation. This adaptive regulation reduces the variance and bias introduced by unreliable or malicious updates, yielding a cleaner descent direction for the global model.

Let us consider the metacognitive function $\mathcal{M}(\mathcal{L}(\theta), \mathcal{A})$. The function monitors the models submitted by participating clients and controls how many models are aggregated by the server over $N$ rounds. Thus, $\mathcal{A}$ represents the set of models submitted by clients participating in the FL process, and $\mathcal{M}(\mathcal{L}(\theta), \mathcal{A})$ selects the best clients' models participating in the aggregation to minimize $\mathcal{L}(\theta)$, exhibiting metacognitive monitoring and control. The process of client selection for aggregation can be formalized as: $\mathcal{M}(\mathcal{L}(\theta), \mathcal{A}) = \mathcal{G} \subseteq \mathcal{A}$, where $\mathcal{G}$ represents clients' models selected for aggregation out of all clients set $\mathcal{A}$. Next, we will show that employing $\mathcal{M}(\cdot)$ provably accelerates model convergence in FL settings.

**Theorem 1 (Convergence Preservation under Metacognitive Monitoring and Control)**: First, we show that introducing metacognition in FL does not violate convergence. Consider the weights of a global model $w_t^{\mathcal{A}}$ composed by the aggregation of all local models in $\mathcal{A}$ and $w_t^{\mathcal{G}}$ composed by the aggregation of models in $\mathcal{G}$ through FedAvg. If $\forall \varepsilon > 0, \exists N_1 \in \mathbb{N}$ s.t. $\forall t \geq N_1, \left\| w_t^{\mathcal{A}} - w^* \right\| < \varepsilon$, then $\exists N_2 \in \mathbb{N}$ s.t. $\forall t \geq N_2, \left\| w_t^{\mathcal{G}} - w^* \right\| < \varepsilon$. That is, assuming the original learning algorithm converges, an algorithm augmented

with metacognitive monitoring and control also converges.

**Discussion:** removing anomalous clients from the FL aggregation does not violate the convergence of the original algorithm if it still converges even under the attacks or anomalies. If the original model $m^{\mathcal{A}}$ does converge, this implies that the attack is not strong enough, which in practice can occur due to various reasons, such as a low proportion of malicious clients or the attack goal was to make it converge to the wrong model. With the convergence of $m^{\mathcal{A}}$, we can guarantee that if the anomaly detection and exclusion is applied, $m^{\mathcal{G}}$ will always converge to the correct model and faster than $m^{\mathcal{A}}$, which is shown in the next part of the theorem. Furthermore, we make a stronger assumption that even if $m^{\mathcal{A}}$ does not converge, $m^{\mathcal{G}}$ will still converge.

**Theorem 2 (Accelerated Convergence under Metacognitive Monitoring and Control)**: If $N_1$ is the round, on which the conventional FL with all clients (no clients removed) converges on $w_t^{\mathcal{A}}$, that is $\forall t \geq N_1, \left\| w_t^{\mathcal{A}} - w^* \right\| < \varepsilon$, and $N_2$ is the round, on which FL with good clients only (bad clients are removed) converges on $w_t^{\mathcal{G}}$, that is $\forall t \geq N_2, \left\| w_t^{\mathcal{G}} - w^* \right\| < \varepsilon$, then $N_2 \leq N_1$.

**Discussion:** the implication of this theorem is that, when using only the updates from clients without outlier updates (as in $w_t^{\mathcal{G}}$), the convergence towards the optimal model will be faster than when aggregating updates from all clients, including those with outlier updates (as in $w_t^{\mathcal{A}}$). This is because the outlier updates, which may significantly deviate from the optimal model, distort the global model, causing it to remain far from an optimal solution for a longer period.

**Theorem 3 (Enhanced Convergence Rate under Metacognitive Monitoring and Control)**: The distances between good models' and optimal model weights is bounded by the distances between all models' and optimal model weights, that is $\exists N \in \mathbb{N}$ s.t. $\forall t \geq N, \left\| w_t^{\mathcal{G}} - w^* \right\| \leq C \left\| w_t^{\mathcal{A}} - w^* \right\|$, where $C$ is a constant if the number of malicious clients does not change during learning and $C = \sqrt{\frac{|\mathcal{G}|}{|\mathcal{A}|}} \leq 1$.

## *IntelliFL* IN PRACTICE: BENCHMARKING AND ANALYZING CASES OF METACOGNITIVE FL APPLICATION DESIGN

In this section, we demonstrate how our framework can be employed to investigate and optimize the FL parameters for user-specified use-cases. The process of the AI-assisted case-study generation and refinement is demonstrated in Table 2. We demonstrate and discuss metrics that were directly obtained during the FL execution, and additionally provide the AI agent reasoning on the collected metrics in order to demonstrate its capabilities in assisting users with the optimization of practical FL deployment configuration.

Within our case study we assess FL setup among two objectives: (1) we select the aggregation algorithm that satisfies either of characteristics such as the fastest convergence of aggregated model, the highest performance of detecting and removing anomalous models from centralized aggregation, and the least computational complexity, and (2), for the selected algorithm, we benchmark how the optimization of its parameters may further enhance FL metacognitive capabilities.

Various aggregation algorithms perform filtering of anomalous model updates differently. Some algorithms are able to immediately react and exclude such models from aggregation; others are unable to exclude all such models due to limitations that are imposed by the algorithm's design and required parameters. These factors result in the delayed convergence of an aggregated model. To address the first objective, we benchmark six aggregation algorithms representing different metacognitive approaches: Trust-based tracking [16], MADE-PI control-based adaptive filtering, distance-based methods (Multi-Krum, Bulyan [17]), statistical robustness (Trimmed Mean [18], [19]), and geometric median (RFA [20]). We set hyperparameters of every algorithm according to theoretical recommendations, which renders the most favorable execution conditions for each of them, allowing the fair comparison under theoretically perfected conditions. Once the optimal algorithm is chosen based on the observed experiment results, in the next experiment session we run five FL executions with the algorithm hyperparameters varied beyond theoretically grounded values to expand the experimental coverage and investigate possible improvements to the algorithm design. As a result, we determine the combination of algorithm hyperparameters that yields the best performance within a given execution environment. This process is straightforward in *IntelliFL*, as it allows to compare FL behavior when executed with a set of varied algorithm hyperparameters. Below we provide a detailed description of results obtained in our case study.

## Case Study: Medical Image Classification

We evaluate FL on the OctMNIST optical coherence tomography retinal imaging dataset using a CNN architecture for the classification task. We execute FL with

TABLE 2: Use case of the application design workflow with *IntelliFL*, as illustrated in Figure 1. This example demonstrates the case of deploying FL in healthcare domain. Based on user-provided high-level description of target system, *IntelliFL* generates and refines case study plan, executes it in FL backend, collects metrics, and provides recommended configuration based on observed FL dynamics.

| Interaction step | Purpose in *IntelliFL* workflow | User input

*IntelliFL* config[b] | AI agent reasoning[a] [15] |
|---|---|---|---|
| **Case study request** | User provides a natural-language FL scenario, describing task, dataset, adversarial conditions, or available devices. If user provides high-level description, the system assumes the best choices that fit this description. | *"We want to deploy FL into a hospital. The facility has older equipment. We plan to collect and label data locally, however, we have some inexperienced personnel who can mislabel some medical samples. We'd like to know which robust aggregation would perform better in this conditions."* | |
| **Initial case study configuration** | System generates the full configuration to be executed in the framework, including dataset choice, model architecture, aggregation strategy, number of clients, training rounds, adversarial attacks or data quality variations. | *Compare client filtering performance of Trimmed Mean (trim ratio = 0.2), RFA (weighted median factor = 1.0), Multi-Krum (number of selections = 7), Bulyan. 10 total clients, 3 of which are anomalous, BreastMNIST dataset, 15 aggregation rounds.* | |
| **AI agent reasoning** | Agent evaluates dataset–model compatibility, round constraints, client availability, hyperparameter defaults, and expected aggregation performance. | | *"This case study simulates label noise with a label flipping attack applied to 30% of clients each round (20% of labels flipped) and compares robust aggregations (Trimmed-Mean, RFA, Multi-Krum, Bulyan) under resource-constrained CPU-only settings."* |
| **Refinement request** | User requests changes such as dataset replacement, adjustment of rounds, attack ratio, or robust aggregation method parameters, if applicable for the particular use-case. | *"Please, make it 20 rounds, with 4 malicious clients, each of which performs 100% label flipping. Test MADE-PI, Trust&Reputation, Multi-Krum, RFA, Bulyan and Trimmed Mean. Use OctMNIST dataset."* | |
| **Final case study configuration** | *IntelliFL* refines the case study configuration based on user request. Number of refinements is unlimited. Once user is satisfied with the case study plan, FL case study is executed and metrics are collected. | *Compare client filtering performance of MADE-PI (p = 1.0, i = 0.05, d = 0.05, num. std. dev = 2.0), Trust&Reputation, Trimmed Mean (trim ratio = 0.2), RFA (weighted median factor = 1.0), Multi-Krum (number of selections = 16), Bulyan. 20 total clients, 4 of which are anomalous, OctMNIST dataset, 20 aggregation rounds.* | |
| **AI agent reasoning** | Reasoning for the updated configuration. | | *"Configured OctMNIST with 20 clients on CPU, 4 fixed malicious clients flipping 100% of labels across all rounds. Compare robust aggregations: MADE-PI, Trust&Reputation, Multi-Krum, RFA, Bulyan, and Trimmed Mean under strict participation."* |
| **Deployment recommendation[c]** | System reasons on the connection of experiment metrics and observed experiment results. | Trust&Reputation with configured parameters as a primary choice, MADE-PI for the simplicity as the deployment hardware is indicated as outdated (lowest loss earlier in the training, computationally inexpensive). | |

[a] Each cell illustrates the exact reasoning saved in the resulting case study configuration file. Generated configuration is a JSON file that includes a full configuration for the case study, including all ML, data heterogeneity, and FL parameters, as illustrated in Figure 2.
[b] Here, for the sake of readability and due to limited space, we present a natural-language description of the major points of generated configuration that best reflect objectives of the user request.
[c] We illustrate simplified interpretation of recommended deployment configuration.

20 aggregation rounds and 20 participating clients, 4 of which flip 100% of labels on their local training subsets, which introduces label-level noise to the model.

Figure 3a shows the loss function of the aggregated

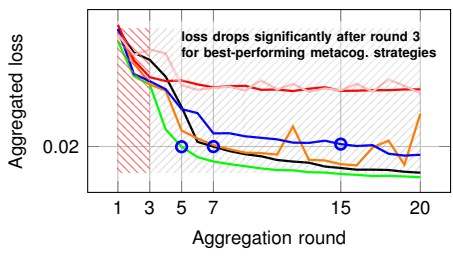

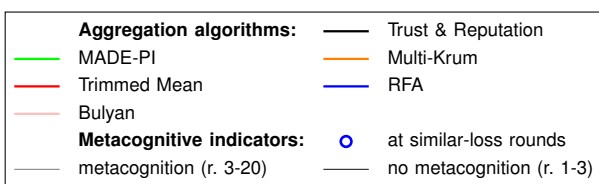

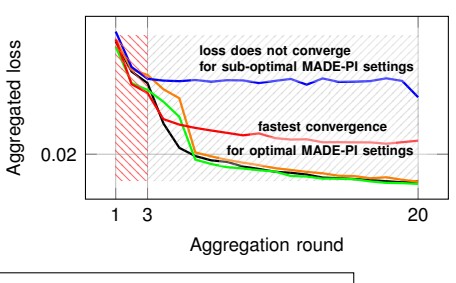

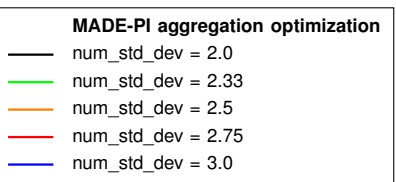

(a) Selection of aggregation algorithm that has the most effective metacognitive monitoring. When algorithm successfully excludes anomalous clients, the loss of the resulting model drops, indicating accelerated convergence. FL with MADE-PI aggregation achieves the lowest loss fastest among all tested algorithms. Multi-Krum and Trust&Reputation achieve similar loss at round 7, RFA at round 15. Others do not converge to the same loss at all.

(b) Selection of optimal hyperparameters for MADE-PI aggregation algorithm. Empirical benchmarking helps to investigate how initial setting of hyperparameters can be further improved, ultimately leading to even faster convergence through better exclusion of anomalous models. Here, the results help to determine reasonable boundaries for the num.std.dev. algorithm hyperparameter, which, in this case, is between 2.0 and 2.5.

FIGURE 3: Selection of FL deployment parameters that render the best *metacognitive monitoring* (convergence acceleration) through the robust aggregation. (a): selection of optimal aggregation algorithm; (b): optimization of selected PID-based aggregation algorithm hyperparameters to further enhance convergence.

model over the course of FL execution. FL executed with Trimmed Mean and Bulyan aggregations resulted in the inability of the centralized model to converge to the same loss values as with other aggregation algorithms. These results can be explained by the performance of the client exclusion which is demonstrated in Table 3, (a). Trimmed Mean has the average exclusion accuracy and precision of 70.00 ± 0.00 and 68.75 ± 0.00, respectively, while Bulyan renders the average exclusion accuracy and precision of 82.67 ± 4.42 and 95.42 ± 2.76, respectively, meaning that at certain aggregation rounds not all of the anomalous clients were excluded. This results in the higher loss values of the aggregated model. Ultimately, these results suggest that some algorithms will render the better metacognitive capabilities in the real-world FL setups, as they are able to better react to environmental conditions resulting in anomalous client-supplied models.

We chose to continue investigation of optimal deployment hyperparameters with MADE-PI algorithm, as it yields the fastest convergence among all algorithms. Results of MADE-PI hyperparameters optimization are demonstrated in Figure 3b and Table 3, (b). Optimization of robust aggregation algorithm hyperparameters to the expected data heterogeneity pattern greatly influences both the speed of convergence of the aggregated model, and the ability of FL aggre-

gation algorithms to adequately react to substandard models. Combining the observations on both the loss over the course of model training and model exclusion performance, one can conclude that MADE-PI aggregation with hyperparameter characterizing number of standard deviations for client exclusion equal to 2.5 results in the best convergence and the most accurate malicious client exclusion, which means that this algorithm with the selected set of hyperparameters allows FL to implement its metacognitive capabilities most effectively.

## Case Study Discussion

This case study demonstrates the practical example of application of our framework. Parameters characterizing expected execution conditions define the FL execution environment. Within this simulated environment, we assessed the performance of robust aggregation algorithms to determine the one that yields the fastest convergence of aggregated model. Then, we explored how this objective can be enhanced even further by investigating the set of hyperparameters that allows one to achieve this result. We demonstrated how *metacognitive monitoring and control* helps to accelerate the learning process. First, out of available aggregation algorithms executed with most favorable configurations, we selected the one that yields the fastest convergence through its metacognitive capabil-

TABLE 3: Assessment of FL *metacognitive control* through the client exclusion performance of robust aggregation methods. More solid green reflects better *metacognitive control* capabilities of the aggregation method, meaning the method detects and excludes anomalous models more accurately.

**(a)** Selection of optimal aggregation algorithm

| Algorithm | Acc. | Prec. | Rec. | F1 |
|---|---|---|---|---|
| T & R | $89.67 \pm 1.25$ | $93.75 \pm 0.00$ | $93.38 \pm 1.38$ | $93.56 \pm 0.71$ |
| MADE-PI | $86.00 \pm 3.27$ | $89.17 \pm 4.25$ | $93.07 \pm 1.32$ | $91.02 \pm 2.29$ |
| M.-Krum | $87.33 \pm 4.42$ | $92.08 \pm 2.76$ | $92.08 \pm 2.76$ | $92.08 \pm 2.76$ |
| Trimm. M. | $70.00 \pm 0.00$ | $68.75 \pm 0.00$ | $91.67 \pm 0.00$ | $78.57 \pm 0.00$ |
| RFA | $81.33 \pm 2.21$ | $94.58 \pm 2.12$ | $84.10 \pm 1.65$ | $89.02 \pm 1.30$ |
| Bulyan | $82.67 \pm 4.42$ | $95.42 \pm 2.76$ | $84.81 \pm 2.46$ | $89.80 \pm 2.60$ |

**(b)** Optimization of MADE-PI hyperparameters

| std. dev | Acc. | Prec. | Rec. | F1 |
|---|---|---|---|---|
| 2.00 | $83.00 \pm 6.53$ | $85.00 \pm 8.16$ | $93.09 \pm 0.63$ | $88.68 \pm 4.78$ |
| 2.33 | $88.00 \pm 2.45$ | $91.67 \pm 3.73$ | $93.26 \pm 1.18$ | $92.40 \pm 1.70$ |
| 2.50 | $89.33 \pm 2.49$ | $93.75 \pm 0.00$ | $93.06 \pm 2.60$ | $93.38 \pm 1.38$ |
| 2.75 | $85.00 \pm 0.00$ | $100.00 \pm 0.00$ | $84.21 \pm 0.00$ | $91.43 \pm 0.00$ |
| 3.00 | $80.00 \pm 0.00$ | $100.00 \pm 0.00$ | $80.00 \pm 0.00$ | $88.89 \pm 0.00$ |

ities. Then, we benchmarked different settings of its hyperparameters to achieve even faster convergence of the resulting ML model in an FL setting, which demonstrates how metacognitive features of FL help to design a more robust end-user application. Finally, the resulting FL configuration is the configuration that is recommended for the target domain under specified execution conditions and data heterogeneity characteristics.

## FRAMEWORK LIMITATIONS

Our framework is implemented to facilitate the design and discovery of optimal FL configuration across a variety of objectives, including enhanced anomaly filtering, accelerated convergence, or lesser computational overhead, depending on the goal of the FL practitioner. While *IntelliFL* provides a platform for investigating an optimal FL setup, it does not provide theoretical guarantees that the resulting FL design will be optimal automatically. Our tool facilitates these objectives by providing an accessible and interactive experience that accounts for challenges inherent to transitioning theoretical FL into real-world systems.

Integration of metacognitive AI agent is limited to generating executable case study configurations based on user-provided natural language descriptions, and analyzing collected metrics with further recommendations. Although the execution is guaranteed with the formal validation module, we did not conduct a study on how accurate is the translation of user requests into executable case study plans. Given that evaluation of LLM response is an emerging problem, we see

the value of our solution in the integration of LLM-generated response into the iterative research and development process, combining traditional software engineering practices for validation with the ability of LLM to analyze provided context to ensure feasibility from the experiment execution perspective.

## CONCLUSION

Our major contributions are fourfold:

**(i)** we introduced *IntelliFL*, a novel design-stage configuration-driven framework that allows FL practitioners to systematically benchmark, compare, and verify different FL procedures and strategies, such as selecting methods that yield greater efficiency, self-adaptation, and robustness under various environmental conditions including adversarial attacks;

**(ii)** we dramatically increased the framework operational intelligence by integrating metacognitive functions and control implemented through the robust aggregation and adaptive parameter selection based on the FL effectiveness and efficiency evaluation by enhancing anomaly filtering, stabilizing model convergence and facilitating context-aware adaptation with generic AI agents;

**(iii)** we proved that application of metacognitive functions and techniques such as self-monitoring,an anomaly detection and exclusion of anomalous causes results in faster learning convergence;

**(iv)** we demonstrated the practical utility of our framework through a series of case studies showing how it guides the design and practical integration of metacognitive AI, allowing the evaluation of capabilities such as self-adaptation and resource rationality by benchmarking aggregation strategy performance and identifying its optimal hyperparameters and deployment configuration. We verified *IntelliFL* flexibility and domain-agnostic nature on multiple data modalities and model architectures, including a CNN for medical image classification and an LLM for medical question-answering.

Our framework significantly increases metacognition usability, since it streamlines the entire experimental process via the single configuration file and the interactive AI agent, dramatically lowering the barriers for users without deep FL expertise. We make our framework publicly accessible to empower the community, accelerate research, and facilitate the metacognitive AI transition from theory to practice.