# OpenReview forum: "Position: Artificial Intelligence Needs Meta Intelligence - the Case for Metacognitive AI"
_ICML.cc/2026/Position_Paper_Track — ICML 2026 Position Paper Track regular_

### Official Review · Reviewer_fL7c · 2026-03-07

**Significance:** 3
**Argument Clarity:** 1
**Rating:** 1
**Confidence:** 4

**Questions:**

None

**Alternative Views Section:**

Yes

**Compliance With Llm Reviewing Policy A Conservative:**

Affirmed.

**Discussion Potential:**

2

**Final Justification:**

Metacognition is an interesting direction worth discussing. Nevertheless, after reading the other rebuttals and replies, I'm even more convinced that the paper needs a major revision. It is not self-contained and overrelies on references to the supplementary paper, giving the impression that it is more about their technical implementation than the position itself.

**Paper Summary:**

The authors state that meta-cognition would be very important for developing future AI systems. In particular, this includes self-monitoring and resource allocation. To this end, the authors advocate for a theoretical framework that unifies different efforts and that allows for the translation of current meta-cognition insights into AI. Furthermore, they discuss research challenges and provide a case study on federated learning based on the IntelliFL framework. The paper ends with a call to action, including further research tasks.

**Position:**

Yes

**Position In Title:**

Yes

**Related Work:**

1

**Strengths And Weaknesses:**

Strengths

* The paper is overall well structured and the authors provide many arguments in line with their position

Weaknesses
* The case study based on IntelliFL is written in a way that it is not fully understandable without reading the IntelliFL paper; the authors actively cite the paper over and over again. Large parts of Section 5 read like an advertisement for IntelliFL and are only indirectly related to the position.
* The authors miss to discuss existing techniques on how the community addresses, e.g., resource allocation. For example, even simple strategies such as early stopping of training are missing. Similarly, more modern approaches, such as a mixture of experts, are not discussed in this context.
* There is quite a connection to AutoML (e.g., cost-aware AutoML) that is also not discussed in the paper.
* How hyperparameters are tuned in their case study is not explained

[Resolved] The alternative view section is missing.

**Support:**

3

---

> ### Author Rebuttal · Authors · 2026-03-31
>
> We thank the reviewer for their feedback and for acknowledging that our paper is well-structured and provides many arguments in line with our position. Please, find our responses to the mentioned weaknesses (W) below.
>
> On the Alternative Views section (W5): The reviewer stated as a weakness that "the alternative view section is missing," and marked "Alternative views section: No" in the form. We respectfully point out that this is factually incorrect. Our paper explicitly includes this required content in Section 3, which is clearly titled "Alternative Views" in accordance with the Position Paper Track instructions. We kindly request that the reviewer re-evaluate the manuscript with this section taken into account.
>
> On the IntelliFL case study (W1): We appreciate the perspective provided on Section 5. We would like to clarify that because this submission is to the Position Paper Track, our primary objective is to advocate for a novel, overarching perspective on AI design rather than to present a traditional empirical systems paper. Consequently, an exhaustive technical description of the IntelliFL framework and its extensive empirical results were deliberately omitted; making them the focal point would have detracted from the broader metacognitive argument we aim to establish. Our intention was strictly to use IntelliFL as an "existence proof" — a practical demonstration of how abstract metacognitive design principles can be operationalized into a functional prototype utilized to design metacognition-aware AI systems. The frequent citations were included to provide interested readers with the underlying empirical evidence and technical architecture that far exceed the page limits and scope of a position paper.
>
> We acknowledge the reviewer's critique that the theoretical connection could be clearer within the text itself. To address this, we kindly refer the reviewer to our response to Reviewer PBRY (specifically Q2). In that response, we have provided a detailed and grounded justification, explicitly mapping how each specific term in our theoretical formal objective (Equation 1) translates directly to the monitoring and control mechanisms within the FL environment. In the revised version of our paper, if accepted, we will integrate this formal mapping to ensure Section 5 is more self-contained, shifting the text from framework-specific descriptions and focusing on the metacognitive principles the case study demonstrates.
>
> On existing techniques and AutoML (W2, W3): We thank the reviewer for highlighting the relevance of early stopping, MoE, and cost-aware AutoML, and we will expand our related work section to integrate these concepts if the paper is accepted. However, we would like to clarify a fundamental distinction between these established efficiency paradigms and the proposed metacognitive framework. Traditional cost-aware AutoML predominantly optimizes training-time configurations to produce a static, frozen model with a fixed efficiency profile. Similarly, while MoE architectures inherently reduce per-token computational overhead through sparse routing, standard mechanisms are designed to maximize model capacity and feature specialization, applying a fixed compute budget per token regardless of task difficulty. In contrast, our metacognitive approach is aimed at reshaping both the training and inference phases. During training time, rather than merely searching for optimal static hyperparameters or routing gates, the model undergoes metacognitive fine-tuning and to explicitly learn how to anticipate task difficulty, estimate the “return over investment” for additional computation, and strategically plan global budget allocations. This training-time metacognition is what empowers the system to perform test-time adaptive computation, actively deciding whether and how much sequential effort to allocate per individual instance. Treating computational effort as a dynamic, autonomous decision variable, executed during active inference remains a distinct and highly specialized frontier compared to AutoML or capacity-driven MoE.
>
> On hyperparameter tuning (W4): We respectfully note that we are not entirely sure which specific hyperparameters are being referenced, whether they are ML model related parameters like learning rate and batch size, or FL related parameters such as the number of aggregation rounds and local rounds. We chose not to include these tuning details in the manuscript because we do not believe they are central to the core purpose of a position paper, which is primarily to state a position, discuss alternative views, and formulate broader research questions and challenges. Additionally, as we describe in the supplementary paper on the IntelliFL framework, our system includes an integrated AI agent that can help users select appropriate hyperparameters based on natural language descriptions and can assist with parameter adjustment and fine-tuning based on the analysis of experimental runs.

---

> > ### Author Rebuttal · Reviewer_fL7c · 2026-04-01
> >
> > Thanks for your rebuttal. Although I like your overall idea, even with the rebuttal (including the reply to Reviewer PBRY), I believe a major revision of the paper is needed, and I cannot approve it based on the rebuttal alone. I see a substantial problem with how the position paper is tightly connected to (and repeatedly refers to) the "supplementary paper on the IntelliFL framework". The position paper is hardly understandable without knowing this supplementary paper. Therefore, I stay with my rating.

---

### Official Review · Reviewer_sRN1 · 2026-03-11

**Significance:** 3
**Argument Clarity:** 3
**Rating:** 4
**Confidence:** 3

**Questions:**

1. The paper places a wide range of capabilities, including monitoring, control, resource allocation, error detection, and adaptation, under the notion of metacognition. Could the authors provide a stricter operational definition and clarify which mechanisms should truly be considered metacognitive, rather than general optimization, control, or robustness methods?

2. The paper argues that metacognitive AI can better balance accuracy, efficiency, and safety. Could the authors make the evaluation criteria more explicit? Specifically, what measurable conditions should a system satisfy to be considered as demonstrating effective metacognition, rather than simply achieving better overall performance?

3. In the case study, one goal is to choose the aggregation algorithm with the fastest convergence, the best anomaly detection performance, and the lowest computational cost. This currently appears closer to a multi-objective engineering selection process. Could the authors clarify what fundamentally distinguishes this from a standard benchmark-based selection pipeline, and why it should be interpreted as metacognitive behavior rather than conventional system optimization?

**Alternative Views Section:**

Yes

**Compliance With Llm Reviewing Policy A Conservative:**

Affirmed.

**Discussion Potential:**

3

**Final Justification:**

The rebuttal improved the conceptual clarity of the submission, particularly in distinguishing meta-level control from standard optimization and in clarifying the intended role of the case study. These points were helpful and addressed part of my earlier concerns regarding framing and interpretation.

However, my main concerns remain only partially resolved. In particular, the rebuttal does not sufficiently establish under what conditions metacognitive mechanisms can reliably improve system performance, nor does it fully clarify how the proposed framework can be operationally distinguished from existing adaptive or multi-objective optimization approaches. The follow-up response was somewhat defensive and did not substantially add new evidence beyond this clarification. While the existence-proof perspective is reasonable for a position paper, it does not fully address the lack of stronger support for the broader claims.

Overall, I appreciate the clarifications and the discussion value of the work. The rebuttal reinforced rather than changed my prior assessment, and I therefore maintain my original borderline accept recommendation.

**Paper Summary:**

This position paper argues that metacognition should be treated as a general design principle for AI systems. The authors contend that modern AI increasingly faces challenges related to computational cost, robustness, and safety during both training and inference, and that simply scaling up computation or relying on post-hoc correction is insufficient. Drawing on metacognitive theories from cognitive science, the paper proposes that AI systems should be equipped with the ability to monitor and regulate their own internal states, error risks, and resource consumption, so as to better balance accuracy, efficiency, and safety. The paper further outlines several open challenges for metacognitive AI and uses federated learning as a case study to illustrate how client trustworthiness estimation, anomaly detection, and dynamic aggregation can be framed as a metacognitive control mechanism.

**Position:**

Yes

**Position In Title:**

Yes

**Related Work:**

3

**Strengths And Weaknesses:**

Strengths:

1. The paper focuses on the trade-off among cost, robustness, and safety in AI systems, which is indeed an important challenge for current large-scale models and learning systems.

2. The paper presents a clear interdisciplinary perspective by linking the notion of metacognition in cognitive science to resource allocation, error monitoring, and control strategies in AI. The overall narrative is coherent and grounded in cognitive science theory.

3. The paper does not stop at a conceptual proposal, but also identifies a number of research challenges and future directions, which may be useful for stimulating broader discussion in the community.

4. The federated learning case study helps reduce the level of abstraction. Although it is more illustrative than conclusive, it still provides a concrete example that helps readers understand how the proposed “monitoring + control” perspective maps onto an actual AI setting.

Weaknesses:

1. Although the paper explicitly discusses metacognition failure and acknowledges that more accurate metacognitive detection does not necessarily lead to better overall system performance, this also reveals that the central claim is still lacking a sufficiently strong positive argument. In other words, the paper raises the right question, but does not yet clearly establish under what conditions metacognitive mechanisms can reliably and predictably improve AI systems.

2. The introduction of the IntelliFL framework makes the proposal feel more actionable, but in its current presentation it mainly shows that these ideas can be instantiated as a tool, rather than demonstrating that the tool validates the distinctive benefits of metacognitive AI over existing system design paradigms. As a result, the practical part remains more illustrative than evidential, and still falls short of supporting stronger general conclusions.

**Support:**

3

---

> ### Author Rebuttal · Authors · 2026-03-31
>
> We thank the reviewer for their thoughtful assessment and for recognizing that our paper addresses a critical challenge in modern AI. We address the weaknesses and questions below.
>
> On conditions in which metacognition can improve AI systems (W1): Regarding the observation that we have not yet established the exact conditions under which metacognitive mechanisms reliably and predictably improve AI systems, we completely agree that this is a critical gap in the current research landscape! This is why we framed this as a position paper and why we identified open Research Questions calling for future work.
>
> Regarding a stricter operational definition (Q1): We define a mechanism as truly metacognitive when it involves a distinct, higher-level supervisory control layer that explicitly monitors and regulates a lower, object-level cognitive process. General optimization and robustness methods typically operate strictly at the object level, permanently altering a model's weights, architecture, or training data to minimize a primary task loss. In contrast, a metacognitive mechanism utilizes a separate, learnable policy to dynamically evaluate its own internal state, such as continuous uncertainty or error risk, and subsequently intervenes by altering processing strategies or resource allocation in the learning process. It is this explicit architectural separation of the task-level execution from the meta-level self-monitoring and control that distinguishes metacognition from conventional, static optimization.
>
> On evaluation criteria (Q2): It depends on what the reviewer means by “simply achieving better overall performance.” As formalized in Equation 1, the training objective for metacognitive systems is – by including a metacognitive control layer – to improve some domain-specific objectives while also minimizing the computational overhead of both the monitoring process and the subsequent control actions. Evaluating a metacognitive system is thus a multi-objective problem, and an “effective” metacognitive controller can indeed be seen as one which increases “overall” performance once additional costs (e.g., compute time, memory usage, resource consumption) are included as part of some overall weighted objective.
>
> Regarding the distinction between our FL case study and a standard multi-objective engineering selection process (Q3): The fundamental difference lies in dynamic, state-dependent adaptation by a metacognitive controller versus static, offline optimization. A standard benchmark-based selection pipeline or multi-objective optimization approach identifies a static configuration prior to deployment based on offline validation. For instance, an engineer might permanently select a robust aggregation algorithm because it balances security and convergence on a hold-out set. In our metacognitive approach, the selection is active and can change moment by moment. The metacognitive controller’s parameters that decide when to change the strategy are themselves trained. This continuous, self-reflective feedback loop acting during execution is the main benefit of metacognitive behavior, distinguishing it from conventional, offline system optimization.

---

> > ### Author Rebuttal · Reviewer_sRN1 · 2026-04-02
> >
> > Thank you for the clarification. The rebuttal helps make the authors’ intended framing clearer, especially the distinction between a meta-level supervisory control layer and standard static optimization, which is more consistent with the paper’s object-level / meta-level discussion.
> >
> > That said, my main concerns are only partially addressed. The response still does not more concretely establish when metacognitive mechanisms can reliably improve AI systems, and the paper itself acknowledges that better metacognitive detection does not necessarily imply better overall system performance. I also still view the FL case study as more illustrative than evidential, since it remains close to algorithm selection and tuning under multiple objectives.
> >
> > Overall, the article explores an important concept, and the rebuttal improves the conceptual clarity of the submission. A pressing question explored by this article is how metacognitive AI can be operationally distinguished and evaluated beyond general adaptive optimization. I remain somewhat unconvinced on that point, but I still think the paper has discussion value as a position paper, so I am comfortable maintaining my original borderline accept recommendation.

---

### Official Review · Reviewer_PBRY · 2026-03-13

**Significance:** 3
**Argument Clarity:** 3
**Rating:** 5
**Confidence:** 4

**Questions:**

1. What can a metacognitive architecture do, in technical terms, that could not be obtained by combining existing tools such as uncertainty estimation, adaptive computation, anomaly detection, and model selection without invoking this broader framing?
2. Could the authors connect the formal objective more directly to the federated learning example by specifying how each term corresponds to the actual monitoring and control mechanisms used in IntelliFL?
3. Is conscious processing important for metacognition? If yes, how can we approach this, given most people would agree that modern LLMs do not model consciousness?
4. The paper presents metacognitive failure as an important open problem. What kinds of failure are specific to this sort of architecture, and how might they differ from failures in systems that do not include a separate supervisory layer?
5. The paper begins by highlighting the practical pressures created by LLMs and other costly AI systems. How do the authors imagine their proposed framework being applied in autoregressive generation or reasoning-heavy inference settings?

**Alternative Views Section:**

Yes

**Compliance With Llm Reviewing Policy A Conservative:**

Affirmed.

**Discussion Potential:**

3

**Final Justification:**

The paper makes a well-supported case for metacognition as a design principle for AI, and it draws on cognitive science more strongly than most position papers in this area. My main concern was the gap between the formal objective and the FL case study. The rebuttal addressed this convincingly by showing how Equation 1 maps onto the FL system, and the authors have said they will incorporate this and expand the related work in the camera-ready version. The question of whether metacognition is truly a new organising principle or instead a reframing of existing methods is still only partly resolved, but this is exactly the kind of useful debate that the position paper track should make space for. I have increased my score from 4 to 5.

**Paper Summary:**

This paper argues that metacognition should be a core principle in new AI system design. The authors define metacognition as a system’s ability to assess its own state, judge uncertainty, and manage its use of computational resources. They support this idea with concepts from cognitive science, such as monitoring and control frameworks, dual-process theories, and resource-rational reasoning. The paper introduces a formal objective that balances performance with the extra cost of adding a metacognitive controller. To illustrate their point, the authors use a federated learning example where a metacognitive mechanism evaluates client reliability and adjusts aggregation. The authors end by suggesting research directions to help make metacognition a broader focus in AI.

**Position:**

Yes

**Position In Title:**

Yes

**Related Work:**

3

**Strengths And Weaknesses:**

I think the paper has several strengths. First, it addresses a quite timely issue. As AI systems become more costly to operate and harder to trust, the need for them to manage their own reasoning and resources is growing. Second, the paper builds on solid work from cognitive science, rather than using metacognition as just a loose idea. This gives the argument more structure than many similar papers I have seen. Third, the formal objective is helpful because it clearly shows the main idea of adding a controller can increase costs, but it may be worth it if it leads to better decisions about computing. Fourth, the federated learning example keeps the paper from being too abstract by giving readers a practical case to consider. Finally, the closing research agenda is well-organised and gives the community a clear place to start.

That being said, however, I am not fully convinced by the current version. The main concern is that the paper does not clearly show what is gained by bringing together existing ideas under the term metacognition. Many of the concepts discussed, like uncertainty estimation, anomaly detection, model selection, adaptive computation, and reflective prompting, already exist in AI and machine learning. The paper notes this, but it does not fully explain what new possibilities arise when these elements are combined in a single metacognitive framework, apart from better terminology or unity. This is important because the paper argues for metacognition as a general design principle, not just as a new way to look at existing methods. I have personally worked on this area too, so I found the core arguments relatively straightforward, but I am not entirely sure this will be the case for another reader.

Another weakness is the gap between the broad motivation and the example chosen. The paper starts by discussing the rising costs and limits of large AI systems, especially those needing expensive inference or complex reasoning. This creates an expectation that the paper will address those situations directly. Instead, the main example is a federated learning case focused on spotting problematic client updates. While this is a valid application, it does not connect as strongly to the initial motivation. Also, the federated learning example is similar to existing work on robust aggregation and anomaly detection, so it is not clear that the metacognitive approach offers a truly new capability rather than just a new way to present known methods.

I also think the formal aspects could be more closely linked to the rest of the paper. The optimisation objective introduced is promising, but it is not clearly connected to the case study. I would have liked to see a direct link between the terms in the equation and the parts of the federated learning system. Without this, the math feels slightly separate from the practical example. Also, the discussion of metacognitive failure is too brief, even though it is a key issue. If these systems are supposed to supervise or control lower-level processes, mistakes at this level could have serious effects. The paper mentions this challenge, but the consequences need more discussion. The related work section could also be improved by engaging more with recent research on adaptive computation, routing, early exit methods, and self-evaluation in language models.

Overall, I think this is a thoughtful and worthwhile position paper, and I found the central idea interesting. The cognitive science foundations are stronger than in many papers of this kind, and the authors do a good job of turning an abstract concept into a recognisable research programme. My hesitation is mainly about scope and distinctiveness. The current draft argues convincingly that metacognitive ideas are relevant to AI, but it is less convincing in showing that metacognition should be elevated to a unifying principle across the field, rather than being viewed as a reframing of several techniques that already exist. I would therefore lean positive though cautious. With a clearer account of what is technically new, a stronger bridge between the general claims and the case study, and a wider engagement with neighbouring literatures, the paper would become much more persuasive.

**Support:**

3

---

> ### Author Rebuttal · Authors · 2026-03-31
>
> Q1: We thank the reviewer for careful reading. We have two replies to this:
>
> First, while certain tools within uncertainty estimation, adaptive computation, anomaly detection, and model selection are highly valuable and may fit within the metacognition framework, they are currently fragmented. We argue that there is value in bringing together disparate sub-disciplines under the single “metacognition” banner with shared research questions and goals.
>
> Second, not all methods within these sub-disciplines fit the metacognitive framework, especially when they are applied post-hoc. Consider the example of LLM post-hoc error correction: current “generate-and-verify” pipelines allow an LLM to make an error and then spend significant, additional resources to fix it after the fact. Post-hoc methods strictly increase computational costs, while a proactive metacognitive approach allocates resources up front and has the potential to correct mistakes before they happen.
>
> In sum, we agree with the reviewer that there are aspects of metacognition in other sub-disciplines and there are aspects of those disciplines in the metacognitive framework, but our position is nonetheless that there is value in identifying a category of metacognitive problems and metacognitive solutions that cut across and reorganize these other domains.
>
> Q2: We will detail the mapping of Equation 1 to our FL case study in the final version, if accepted. In standard FL, minimizing expected loss to find $\theta^* = \arg\min_\theta\mathcal{L}(\theta)$ relies on the static aggregation of local updates $\theta_t^{(k)}$ at round $t$, such as FedAvg where $\theta_{t+1} = \sum_{k=1}^K\frac{n_k}{D}\theta_t^{(k)}$. In our framework, the task-level function $f_w$ translates to this aggregation process, dynamically parameterized by a mode $c \in \mathcal{C}$ representing the chosen strategy (e.g., FedAvg, a robust algorithm, or a trusted client subset $\mathcal{G}$). The metacognitive policy $\pi_{w'}$ utilizes learnable weights $w'$ to select mode $c$ based on the self-monitoring state $s$. This resource-rational behavior is governed by $cost(c)$, the penalty for executing mode $c$ (e.g. expensive robust aggregation improves task loss $\mathcal{L}(\theta)$ but raises cost), and $||\pi_{w'}||$, the computational overhead of the controller. Because training-time overhead is amortized over the model operational lifetime, an expensive meta-control operation at learning time is far more tolerable than equivalent inference overhead. Minimizing this combined objective creates an adaptive intelligence that dynamically balances these trade-offs, distinguishing it from static optimization.
>
> Q3: We are agnostic to whether or not conscious processing is involved; we approach metacognition more from a functional and computational perspective rather than a phenomenal one. In the absence of an established formal definition of consciousness (a notoriously contentious topic and outside the scope of our position paper), we cannot speak to whether self-monitoring metacognitive systems are or are not using such mechanisms. We also note that in cognitive science, some automatic metacognitive processes, such as Feeling of Rightness cues, seem to operate unconsciously.
>
> Q4: Unlike standard object-level errors, metacognitive architectures introduce a distinct class of failures where the supervisory control layer misjudges its own competence or selects flawed interventions, making the mitigation of these compounding errors a critical open research challenge. In the FL example, this type of failure manifests when the controller selects an inappropriate aggregation strategy, failing to properly balance resource costs against security or efficiency needs. These suboptimal meta-level decisions lead to an increase in the overall loss of the metacognitive policy objective and can destabilize the underlying learning process.
>
> Q5: To address the massive inference costs of modern reasoning models, many current systems employ piecemeal strategies like adaptive routing and tool delegation based on continuous uncertainty evaluation. We argue that unifying these fragmented efforts under a single "metacognitive control" banner provides significant value to the research community. Doing so will establish a shared framework to identify and solve open challenges in AI system design.
>
> While the metacognitive mechanisms operate during inference, we specifically position our proposed IntelliFL framework as a design-stage simulation tool that allows researchers to systematically prototype and evaluate meta-level policies and thresholds before deployment, thereby avoiding the prohibitive costs of testing uncalibrated controllers in live settings. Additionally, we agree with the reviewer’s feedback regarding the related work section and will expand our discussion in the camera-ready version to better engage with the neighboring literature on adaptive computation, routing, and early exit methods.

---

> > ### Author Rebuttal · Reviewer_PBRY · 2026-04-02
> >
> > Thank you for the thorough rebuttal (given the conference constraints). The detailed explanation of how Equation 1 maps onto the FL case study in Q2 was especially helpful and addressed what I saw as the biggest gap in the original manuscript. I would strongly encourage the authors to include this in the camera-ready version. The practical stance on consciousness in Q3 is reasonable, and the clarification around metacognitive failure modes in Q4 is sufficient for a position paper, although this is still an area that would clearly benefit from deeper discussion in future work.
> >
> > I am increasing my score from 4 to 5. The rebuttal has improved the conceptual clarity of the submission, and I now feel more confident that the paper will lead to productive discussion at the conference. My remaining hesitation is that the distinctiveness of metacognition as a unifying design principle, rather than simply a reframing of existing adaptive and resource-aware methods, could still be expressed more sharply. Even so, I recognise that encouraging exactly this kind of debate is part of what makes a strong position paper.

---

### Official Review · Reviewer_R8nY · 2026-03-16

**Significance:** 2
**Argument Clarity:** 2
**Rating:** 4
**Confidence:** 3

**Questions:**

See weaknesses

**Alternative Views Section:**

Yes

**Compliance With Llm Reviewing Policy A Conservative:**

Affirmed.

**Discussion Potential:**

2

**Final Justification:**

Thanks to the authors, and my concerns have been solved.

**Paper Summary:**

Authors advocates for metacognition as a design principle for resource-rational AI, demonstrating its deployment via a Federated Learning case study and introducing a framework for developing metacognition-enabled AI applications.

**Position:**

Yes

**Position In Title:**

Yes

**Related Work:**

2

**Strengths And Weaknesses:**

Strengths:

1. The idea that metacognition as a design principle for resource-rational AI, is interesting.

2. Demonstration on Federated Learning case study strengthens the authors' claim.

Weaknesses:

1. The authors provide two examples on FL and medical image classification, which are limited. I'm very interested in how this idea could be implemented on training LLM.

2. It seems the metacognition is highly dependent on domain knowledge, so how can this method be scaled up?  When this idea been extended to broad domains, does metacognition need human labeling or can be learned thru some learning process?

**Support:**

2

---

> ### Author Rebuttal · Authors · 2026-03-31
>
> We sincerely thank the reviewer for their feedback and for recognizing that the idea of metacognition as a design principle is interesting. We also appreciate the acknowledgement that our FL case study strengthens our claims. We address the specific weaknesses (W) mentioned by the reviewer below.
>
> On LLM implementation and limited examples (W1): We agree with the reviewer that our example case-studies are limited in scope; however, the role of a position paper is not to comprehensively survey but to argue for what should be done. We chose FL examples because the role of metacognitive strategies in learning is especially under-appreciated in the broader community, and we are able to refer to an existing tool with concrete benefits for specific types of AI systems (FL) that supports their design and evaluation before deployment. In fact, the FL tools we cite do already have integrated LLM model architectures and related datasets, including BiomedBERT (PubMedBERT) and GPT-2 models, alongside MedQuad and NER datasets. Due to double-blind anonymity rules, we cannot share the repository or links to any accepted/published papers in this stage, but below we provide anonymous links to figures showing our recent experimental results evaluating these models in IntelliFL framework and some implementation evidence. While the figures may look outside of the context (according to the ICML’s rebuttal policy, we are allowed to provide links only to figures with the caption, no other content), they are serving as proof that the LLMs and their training is already implemented and functional in the cited FL tool.
>
> Figure 1. Performance metrics collected with our frameworks on the empirical study on training LLMs in a federated manner with compromised clients: https://i.postimg.cc/34pbLHfB/Screenshot-2026-03-30-at-6-05-44-PM.png
>
> Figure 2. The up-to-date list of the datasets that IntelliFL allows to experiment with: https://i.postimg.cc/DJsYCKp6/Screenshot-2026-03-30-at-5-53-34-PM.png
>
> Figure 3. The list of models implemented for each of the datasets that IntelliFL allows to experiment with: https://i.postimg.cc/CBD6vVrs/Screenshot-2026-03-30-at-5-47-28-PM.png
>
>
> Regarding the reviewer's interest in how this idea could be implemented on training LLMs: we advocate that metacognitive design inherently generalizes to all AI systems and models, regardless of the underlying architecture (whether LLM, CNN, or otherwise). AI systems must possess a meta-control level allowing them to be resource-rational and make decisions according to the trade-offs specified by the metacognitive control function. We advocate that this principle must be imperative not only for the inference stage, but the learning stage of AI models too, which distinguishes our work from others in the field. This is a high-level control independent of the specific model. Practically, we demonstrate how this idea translates to the learning stage using FL use case as a demonstration — specifically through selecting clients for aggregation, prioritizing certain updates, and dynamically selecting aggregation strategies. We refer the reviewer to the links to the figures posted above. An LLM training pipeline could similarly use a metacognitive controller to dynamically route data batches or adjust compute allocation.
>
> On domain knowledge dependence and scaling (W2): metacognition requires establishing a metacognitive policy, $\pi_{w'}$, which possesses its own learnable parameters, $w'$. This policy efficiently selects a mode of handling an instance, $c$, based on the self-monitoring state, $s$, and/or the data itself. This is defined by Equation 1 in our paper, which explicitly accounts for the added cost of the metacognitive controller itself, $||\pi_{w'}||$. The reviewer is correct that such costs will be domain-specific, but the core idea of learning a meta-policy according to some metacognitive cues is itself universal.
>
> While the metacognitive learning framework itself is rather universal, whether or not it works well in a given domain is a different question. We can take inspiration from Cognitive Science and implement metacognitive signals like the Feeling of Knowing, but these are generally considered to be heuristics without guarantees across arbitrary domains. Which metacognitive cues are effective in which domains (and why) is a problem requiring further research; this relates to the Research Challenges we identified in Section 4 of our position paper.

---

> > ### Author Rebuttal · Reviewer_R8nY · 2026-04-01
> >
> > Thank you for your response, but I can't open the images in the link

---

### Decision · Program_Chairs · 2026-04-30

**Decision:**

Accept (regular)

**Comment:**

Most reviewers praised the paper for the unique and insightful core position, with some reviewers especially appreciating the timeliness of the paper as well as the interdisciplary inspiration it draws from cognitive science. Reviewers further appreciate the actionable nature of the paper, despite being a position paper. One reviewer in particular (fL7c) argued in in favor of rejecting the paper, though on my reading the criticism is outweighed by the positive appraisal from the other reviewers.

In light of this I suggest recommend acceptance.